# Understanding the genetic determinants of the brain with MOSTest

Dennis van der Meer [1,2,9 ✉], Oleksandr Frei [1,3,9], Tobias Kaufmann [1], Alexey A. Shadrin[1], Anna Devor [1,4,5], Olav B. Smeland [1], Wesley K. Thompson[1,6], Chun Chieh Fan [7], Dominic Holland[4,7], Lars T. Westlye [1,8], Ole A. Andreassen [1] & Anders M. Dale[1,7 ✉]

Regional brain morphology has a complex genetic architecture, consisting of many common polymorphisms with small individual effects. This has proven challenging for genome-wide association studies (GWAS). Due to the distributed nature of genetic signal across brain regions, multivariate analysis of regional measures may enhance discovery of genetic variants. Current multivariate approaches to GWAS are ill-suited for complex, large-scale data of this kind. Here, we introduce the Multivariate Omnibus Statistical Test (MOSTest), with an efficient computational design enabling rapid and reliable inference, and apply it to 171 regional brain morphology measures from 26,502 UK Biobank participants. At the conventional genome-wide significance threshold of $\alpha = 5 \times 10^{-8}$, MOSTest identifies 347 genomic loci associated with regional brain morphology, more than any previous study, improving upon the discovery of established GWAS approaches more than threefold. Our findings implicate more than 5% of all protein-coding genes and provide evidence for gene sets involved in neuron development and differentiation.

[1] NORMENT Centre, Division of Mental Health and Addiction, Oslo University Hospital & Institute of Clinical Medicine, University of Oslo, Oslo, Norway. [2] School of Mental Health and Neuroscience, Faculty of Health, Medicine and Life Sciences, Maastricht University, Maastricht, The Netherlands. [3] Center for Bioinformatics, Department of Informatics, University of Oslo, Oslo, Norway. [4] Departments of Neurosciences and Radiology, University of California at San Diego, La Jolla, CA 92037, USA. [5] Martinos Center for Biomedical Imaging, MGH/HMS, Charlestown, MA 02129, USA. [6] Department of Family Medicine and Public Health, University of California at San Diego, La Jolla, CA 92037, USA. [7] Center for Multimodal Imaging and Genetics, University of California at San Diego, La Jolla, CA 92037, USA. [8] Department of Psychology, University of Oslo, Oslo, Norway. [9] These authors contributed equally: Dennis van der Meer, Oleksandr Frei. ✉email: d.v.d.meer@medisin.uio.no; andersmdale@gmail.com

Regional surface area and thickness of the cerebral cortex, and volume of subcortical structures, are highly heritable brain morphological features with complex genetic architectures, involving many common genetic variants with small effect sizes[1,2]. The predominant strategy for identifying genomic loci associated with complex traits is through genome-wide association studies (GWAS), a mass-univariate approach whereby the association between a single outcome measure and each of millions of genetic variants, in isolation, is tested. This is accompanied by a stringent multiple comparison correction to control the family-wise error rate, necessitating very large sample sizes to identify even relatively strong effects. To date, the largest GWAS of regional brain morphological features, based on brain scans obtained from up to 50,000 individuals, identified almost 200 genetic variants[1], which together explained only a fraction of the reported narrow-sense heritability. These studies primarily investigate each region of interest individually, compounding the multiple-comparisons correction problem.

In addition to small effect sizes across many variants, the genetic architectures of sets of regional brain features are likely to strongly overlap. Gene expression studies of the human brain have shown widespread gradients across the cortex[3]. Thus, genetic variants probably have distributed effects across regions and morphological measures. We have shown that cortical thickness and surface area have extensive genetic overlap[4], despite reports that they are phenotypically and genetically only weakly correlated to each other[5], due to mixed directions of effects of the underlying genetic variants[4,6]. The discovery of these variants may be boosted through joint analysis of these traits in a multivariate framework. This avoids the family-wise error rate penalty for studying multiple outcome measures, or the use of strategies that reduce phenotypic information to a single composite score, which can cause considerable loss of statistical power[7]. Importantly, a multivariate approach is much more consistent with the notion of the brain being an integrated unit, with highly interconnected and biologically similar brain regions, compared to univariate approaches that ignore the information shared across these component measures.

Several multivariate approaches to GWAS have been proposed to date[8,9]. In this context, multivariate association at a given single-nucleotide polymorphism (SNP) means that at least one of multiple traits being considered is associated with the genotype vector of that SNP[9]. To test for multivariate association, MQFAM (also known as MV-PLINK)[10] and MultiPhen[11] both perform a multiple regression whereby the genotype vector is used as an outcome variable, while each phenotype is turned into an explanatory variable; the p-value is then calculated from an F-test, which tests for an association between the genotype vector and the most predictive linear combination of phenotypes at each SNP. The advantage of MultiPhen is that it uses ordinal regression, whereas MQFAM is based on canonical correlation analysis, which in theory is less appropriate for prediction of a categorical variable such as 0-1-2 coded genotype vector. The statistical power of both methods is known to be similar to MANOVA[8]. MultiABEL is another multivariate GWAS approach, which implements Pillai's trace MANOVA[12] to calculate multivariate p-value from summary statistics; its authors further advocate the importance of rank-based inverse-normal transformation. Most recently, aMAT[13] was introduced. This test, based on a chi-square test statistic, explores regularization (spectral filtering) of the correlation matrix R as a way to further boost statistical power in multivariate methods. Distinct from these multivariate tests of association is the multi-trait analysis of GWAS (MTAG) method[14], which boosts discovery in a primary trait by conditioning on one or more secondary traits and therefore does not test the multivariate null hypothesis that none of the traits are associated with a given SNP. Although all multivariate methods listed above have been shown to substantially increase gene discovery compared to univariate approaches[8], they have not been widely adopted by large-scale international consortia. This may be due to that fact that the results can be less straightforward to interpret, in addition to high computational costs, lack of user friendliness, lack of method validation, and/or model assumptions not fitting with the real data.

Here we introduce the Multivariate Omnibus Statistical Test (MOSTest), designed to boost the statistical power of imaging genetics by capitalizing on the distributed nature of genetic influences across brain regions and pleiotropy across imaging modalities. MOSTest is efficient and capable of combining large-scale genome-wide analyses of dozens of measures for tens of thousands of individuals within hours while achieving enhanced statistical power. Key steps of the MOSTest analysis include: (1) applying a rank-based inverse-normal transformation to the input measures; (2) estimating the multivariate correlation structure from the GWAS on randomly permuted genotype data; (3) calculating the Mahalanobis norm, as the sum of squared decorrelated z-values across univariate GWAS summary statistics, to integrate effects across the measures into a multivariate test statistic; and (4) employing the gamma cumulative density function to fit an analytic form for the null distribution, enabling extrapolation to and beyond the $5 \times 10^{-8}$ significance threshold. The "Methods" section contains a detailed description of these steps. We further provide extensive simulations of MOSTest performance under a wide range of conditions to validate its assumptions and compare it to other multivariate approaches.

We compare MOSTest with an established inferential methodology recently used by the Enhancing NeuroImaging Genetics through Meta-Analysis (ENIGMA) consortium[1], referred to as the min-P approach; this approach takes the smallest p-value of each SNP across multiple univariate GWAS, and corrects this for the effective number of traits studied[11,15], i.e., shared genetic architecture across traits does not contribute to statistical power. Min-P achieves its maximum power when the genetic effects across traits are independent; conversely, multivariate approaches have greater power when genetic effects are shared across traits[8]. We applied MOSTest to sets of regional brain morphology measures, hypothesizing that it will outperform the min-P approach due to the distributed nature of genetic effects and the presence of pleiotropy across modalities. We find that MOSTest improves locus discovery compared to min-P more than threefold, doubling the effective sample size, with significant loci having effects across regions and across feature subsets. Further, through simulations, we confirm that MOSTest maintains correct type-I error in a range of scenarios. We conclude that, due to the distributed nature of the genetic signal across brain regions, joint analysis of regional morphology measures in a multivariate statistical framework provides a way to enhance discovery of genetic variants with current sample sizes.

## Results

**Locus discovery**. Through MOSTest, we identified thousands of independent SNPs reaching the genome-wide significance threshold of $\alpha = 5 \times 10^{-8}$ across hundreds of independent loci, as shown in Fig. 1a. Overall, MOSTest led to a threefold higher discovery than the min-P approach. The difference in performance is particularly pronounced when all features are combined, as is also evident from the Miami plots shown in Fig. 1b–e. For all features combined, 92 loci were discovered by both MOSTest and min-P, 20 were unique to min-P, and 255 were only discovered by MOSTest. Supplementary Data 1–3 lists these loci, together with for how many and which regions the lead SNPs were

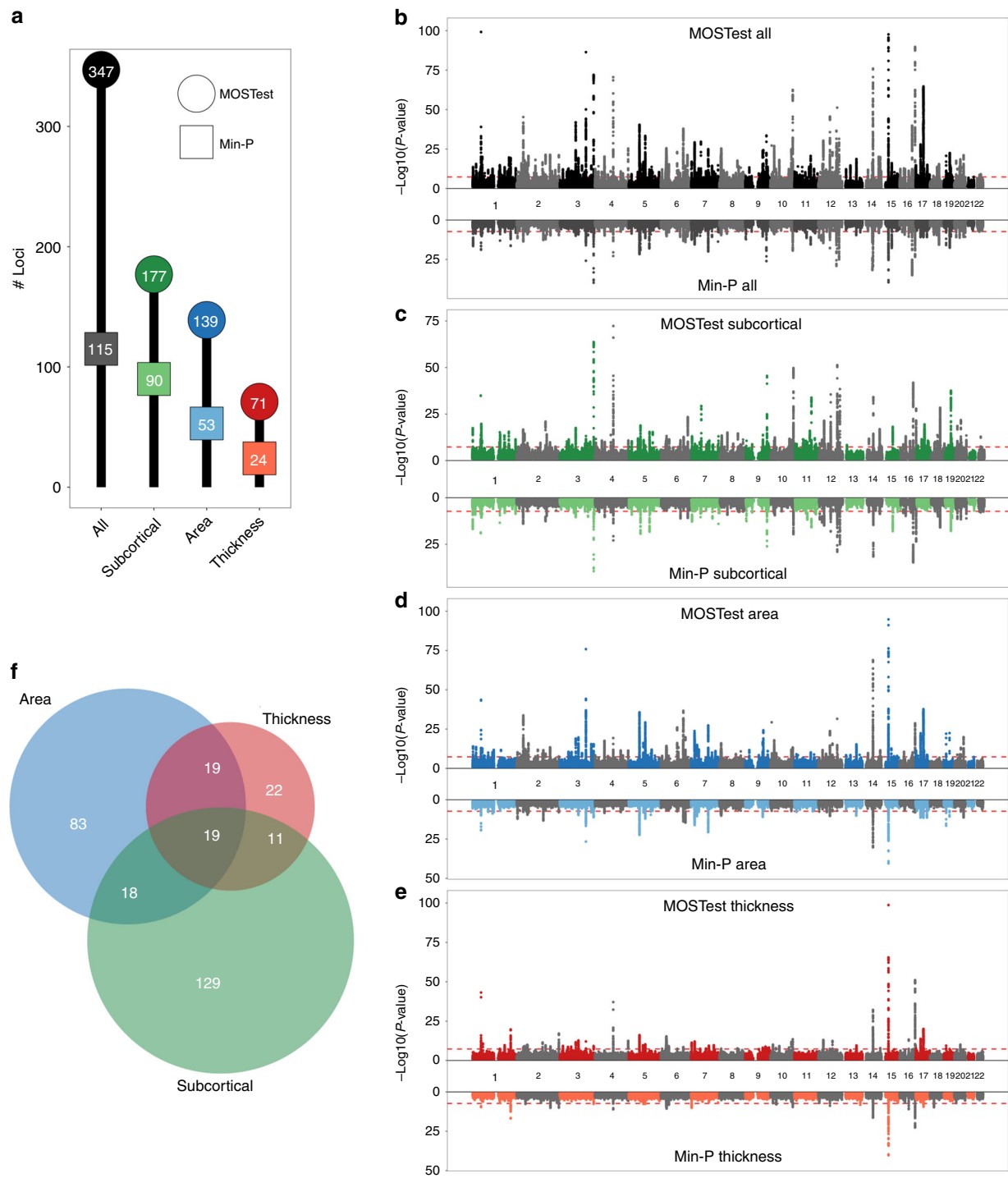

**Fig. 1 MOSTest improves locus discovery. a** Number of independent genome-wide significant loci identified (on the *y* axis and in the bubbles) for each set of features (on the *x* axis), by MOSTest (in darker colored circles) and by min-P (in lighter colored squares). **b**–**e** Miami plots, contrasting the observed $-\log_{10}(p\text{-values})$, shown on the *y* axis, of each SNP for MOSTest (top half) with min-P (bottom half), for each of the feature sets. The *x* axis shows the relative genomic location, grouped by chromosome, and the red dashed lines indicate the genome-wide significance threshold of $5 \times 10^{-8}$. Note, *y* axis is clipped at $-\log_{10}(p\text{-value}) = 100$. **f** Venn diagram depicting the number of loci, identified with MOSTest, overlapping between the three feature subsets.

significant. Generally, loci discovered by both tests had significant effects on multiple regions, loci discovered only by min-P had effects on 1 or 2 regions, and loci discovered only by MOSTest often had no whole-genome significant effect on any of the regions.

As can be seen in Fig. 1f, many loci identified with MOSTest were shared across the three feature subsets. Further, the number of MOSTest-discovered loci for all features combined (347) is larger than the summation of number of unique loci found when analyzing these feature subsets individually (301). This illustrates how MOSTest capitalizes on the distributed and non-sparse nature of genetic effects through the combination of features.

**Replication**. We carried out replication analyses of the GWAS results in an additional 4884 UKB participants, whose neuroimaging

data were released after we carried out our primary analyses. The loci discovered through MOSTest and min-P replicated at similar levels, with ~40% being significant in this smaller additional sample (Table 1). Therefore, the absolute number of loci replicating is three times higher for MOSTest compared to min-P. Supplementary Data 1–3 also lists the replication p-values by both min-P and MOSTest per discovered locus.

**Power**. Using the MiXeR tool[6,16], we fitted a Gaussian mixture model of the null and non-null effects to the GWAS summary statistics, estimating for each feature set the number of SNPs involved, i.e., their combined polygenicity, and their effect size variance, or "discoverability". Please see the "Methods" section for more details. The results are summarized in Fig. 2, depicting the estimated proportion of genetic variance explained by discovered SNPs by both approaches as a function of sample size. The horizontal shift of the curve indicates that the effective sample size of MOSTest is generally about twice as high as that of min-P, with the highest discovery for MOSTest when all features

are combined, and lowest discovery for the set of cortical thickness features.

**Validation and comparison between MOSTest and other tools**. We performed extensive validation of MOSTest methodology and implementation, checking its performance under a range of conditions through simulations with synthetic data, and comparing this with other multivariate approaches besides min-P, namely MultiABEL, MultiPhen, and MQFAM. We used a framework whereby effects are simulated on chromosome 21 to compute statistical power, while all other chromosomes are kept free of genetic signal to estimate type-I error under the null. Supplementary Tables 4 and 5 list the full set of simulation scenarios, such as varying the sparsity of genetic effects and the number of features included, as well as the heritability of these features and their correlation structure. Further details about the simulation framework are provided in the "Methods" section.

Under a range of conditions, all methods showed similar statistical power, except for min-P with lower power. The methods all maintained correct type-I equally well under the null and following permutation, under these conditions, as summarized in Supplementary Figures 1 and 2. However, we could not explicitly validate type-I error for MultiPhen and MQFAM across the genome due to slow runtime; these typically exceeded one hour to calculate p-values per $M = 100$ variants in the power analyses, for each one of 760 simulation runs. Therefore, it was impractical to run these tests for $M = 7.3$ million causal variants. See Supplementary Table 5 for the runtimes per simulation.

The min-P approach outperformed the multivariate methods under one specific condition: when a small set of heritable features are analyzed together with a much larger set of non-heritable features (Supplementary Fig. 3). All tests have similar power when all features are heritable, but do not share genetic variants, or when shared genetic effect sizes follow a heavy-tailed distribution (Supplementary Fig. 4).

The simulations revealed the importance of the rank-based inverse-normal transformation: without this transformation, the tests had inflated type-I error, as well as lower statistical power, when the features were not normally distributed (Supplementary Fig. 5). We note that an incorrect type-I error fully prohibits an application of a statistical test. The MOSTest genotype

---

**Table 1 Equal replication rates of genome-wide significant loci by MOSTest and min-P.**

| Test | Feature set | # loci discovered | # loci replicated | Fraction replicated |
|------|-------------|-------------------|-------------------|---------------------|
| MOSTest | All | 347 | 122 | 0.35 |
| MOSTest | Subcortical | 177 | 68 | 0.38 |
| MOSTest | Surface area | 139 | 58 | 0.42 |
| MOSTest | Cortical thickness | 71 | 24 | 0.34 |
| min-P | All | 115 | 48 | 0.42 |
| min-P | Subcortical | 90 | 40 | 0.44 |
| min-P | Surface area | 53 | 24 | 0.45 |
| min-P | Cortical thickness | 24 | 9 | 0.38 |

Results from the replication of the genome-wide significant loci are identified through MOSTest and min-P in an additional sample of 4884 individuals. The column "# loci replicated" indicates the amount of discovered loci in the main analysis that surpass a nominal significance ($p = 0.05$) in the replication sample. The "fraction replicated" column divides the number of loci discovered in the main analysis with the number of loci that replicated.

---

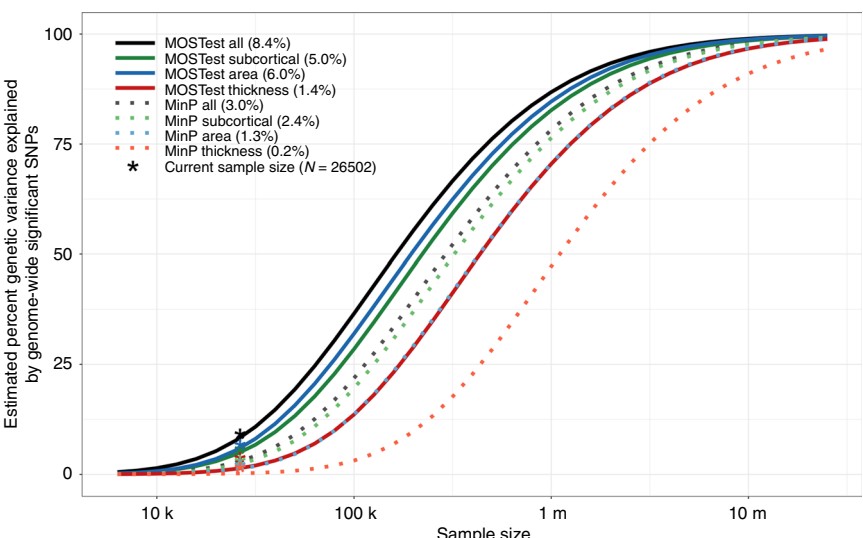

**Fig. 2 MOSTest increases effective sample size.** Estimated percent of genetic variance explained by SNPs surpassing the genome-wide significance threshold, on the y axis, as a function of sample size, depicted on the x axis on a $\log_{10}$ scale, for each of the feature sets and for both MOSTest and min-P methods. Percentages of genetic variance explained by discovered SNPs with current sample size ($N = 26,502$) are shown in parentheses.

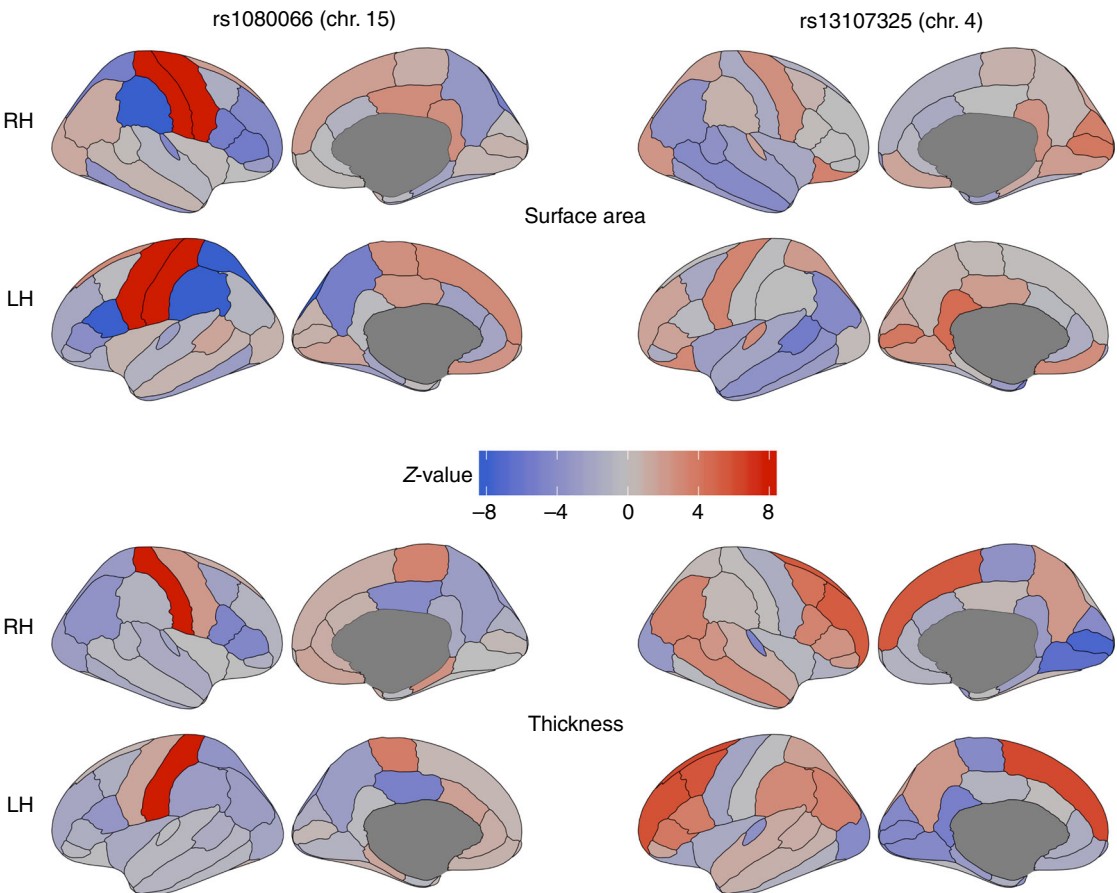

**Fig. 3 The genetic variants identified with MOSTest have distributed effects across the cortex.** Z-values from the univariate GWAS for each cortical region for the two most significant lead SNPs from MOSTest applied to all features combined (left two columns for rs1080066 on chromosome 15, and right two columns for rs13107325 on chromosome 4). The top two rows show the effects of the SNPs on regional surface area, and the bottom two on cortical thickness. Positive effects of carrying the minor allele are shown in red, and negative in blue. Note: the absolute Z-value scaling is clipped at 8 ($p = 1.2 \times 10^{-15}$); an absolute Z-value of 5.45 corresponds to two-tailed genome-wide significance ($p = 5 \times 10^{-8}$). RH = right hemisphere, LH = left hemisphere.

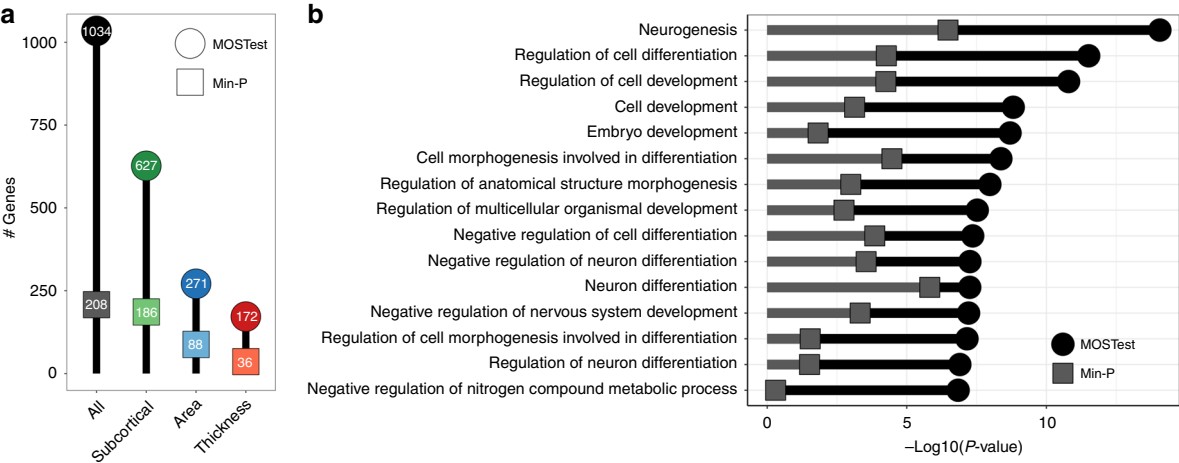

**Fig. 4 Functional mapping and annotation indicates high neurobiological relevance. a** Number of genome-wide significant genes identified (on the y axis and in the bubbles) for each set of features (on the x axis), by MOSTest (in darker colored circles) and by min-P (in lighter colored squares). **b** Results from the gene-set analyses following the application of MOSTest and min-P on all brain features. The 15 most significant Gene Ontology sets for MOSTest are listed on the y axis and −log10(p-values) are shown on the x axis. Multiple-comparisons corrected MOSTest gene-set p-values are indicated by black circles, min-P in gray squares.

permutation scheme, absent in other tools, effectively captures this issue, therefore forms an important part of the test that prevents reporting inflated p-values and spurious associations. Further, Supplementary Fig. 5 shows the role of covariance structure across genetic effects, and across features. We observe that the highest power to detect associations occurs when features have realistic covariance structure, but genetic effects are not correlated.

Our simulations further show that spectral regularization, implemented only in MOSTest, is essential when there is linear dependence between features. Regularization was not necessary in our main analysis, as the conditioning number of phenotypic correlation matrix R was reasonably low (Supplementary Table 4), leading to a well-defined matrix $R^{-1}$. However, in the presence of linear dependence, MOSTest has invalid type-I error without spectral regularization, while correct regularization solves this issue and at the same time improves statistical power (Supplementary Fig. 6). Further, Supplementary Fig. 6 shows that estimation of phenotype covariance matrix can be done either from the phenotypes themselves or from z-scores under permutation, and that these two approaches yield nearly identical results in the simulation scenarios. Note that the permutation scheme has the advantage of not requiring availability of all phenotypes at one site, allowing for application in a meta-analytical setting.

Please see the "Methods" section and Supplementary Information for further validation data, including Supplementary Fig. 7, displaying scaled-down version of the simulations, and Supplementary Fig. 8, QQ plots showing that the MiXeR model correctly captures the LD dependence of the MOSTest association statistics. We also performed LD score regression and used the intercept as an indicator that MOSTest results are free of genomic inflation (Supplementary Table 6). Note that the LD score regression results should be interpreted with care[17], and that we have not provided formal proof that MOSTest p-values scales with LD structure as required by LD score regression model.

**Regional effects**. Cortical maps, depicting the morphological associations of the lead SNPs identified with MOSTest on all features with regional surface area and thickness measures, made clear that these SNPs have distributed effects, often with mixed directions, across regions and feature sets. As an example, Fig. 3 shows the maps for the top two hits (rs1080066 on chromosome 15, $p = 1.2 \times 10^{-305}$, and rs13107325 on chromosome 4, $p = 3.1 \times 10^{-124}$); all other maps are provided in the Supplementary Data. These maps revealed anterior-posterior gradients as well as hemisphere-specific effects of some of the lead SNPs, in line with previously reported genetic patterns of developmental regionalization in the brain[18,19].

Gene-level analyses, using Multi-marker Analysis of GenoMic Annotation (MAGMA)[20,21], indicated that 1034 out of all 18,775 protein-coding genes (i.e., 5.5%) were significant, with a p-value below a Bonferroni-corrected threshold of $\alpha = 0.05/18,775$. Figure 4a shows the number of significant genes for each set of features. Through competitive gene-set analyses, we identified 136 significant Gene Ontology sets for MOSTest applied to all features, the vast majority of which related to neuronal development and differentiation, with Fig. 4b listing the top 15. Please see Supplementary Fig. 9 for the overlap between these pathways, and Supplementary Data 12 and 13 for further information on all significant genes and genetic pathways.

## Discussion

Applying the MOSTest approach to structural brain imaging data, we discovered more loci associated with regional cortical and subcortical morphology than any previous GWAS of brain morphology, even those that had nearly double the sample size[1,2]. Further, a direct comparison with the established min-P method in the same sample revealed a threefold increase in discovery. This improvement indicates the presence of extensive shared genetic architecture across brain regions and across morphological measures, attesting to the importance of estimating levels of polygenic overlap beyond those indicated by genetic correlations[4,6], and arguing for techniques that boost discovery of genetic determinants leveraging shared signal between traits[22]. Indeed, overlapping genetic determinants are to be expected given the shared genetic control of neurodevelopment across brain regions[23], and that similar molecular mechanisms operate across regional borders defined by gross morphological features. This is in accordance with the high levels of pleiotropy across many brain-related traits and disorders[24]. Therefore, our multivariate strategy is better tailored to the underlying neurobiological processes than conventional univariate approaches, as confirmed by our identification of highly significant links to gene sets of neuronal development and differentiation.

Our extensive validation of the MOSTest methodology and implementation show that the MOSTest is a valid statistical test with good statistical power to detect associations in a multivariate context, across a wide range of conditions, suitable for applications to large-scale data with many outcome measures. Key advantages of MOSTest include (1) an orders of magnitude shorter runtime as compared to MQFAM and MultiPhen, (2) a built-in genotype permutation scheme that allows detection of cases of invalid type-I error, (3) the ability to incorporate spectral regularization of the phenotype matrix, and (4) extensive validation in simulations.

Our simulations showed that in some scenarios min-P outperforms the multivariate methods, specifically when genetic signal is sparse across phenotypes. As the level of sparsity will be different across SNPs, it is expected that min-P may discover a few additional SNPs that are missed by MOSTest, as also indicated in our comparison using real data. Further, these simulations made clear it is important to exclude traits with too low heritability to achieve best possible power. We are planning to develop an automated regularization strategy, in line with the aMAT method[13], to select the best possible trade-off between univariate and current multivariate methods, to further improve power of the multivariate analysis. Further, due to practical limitations, we performed simulations of up to a hundred features, leaving validation of the MOSTest procedure when including more features as future work. Enabling the MOSTest procedure, and particularly its genotype permutation scheme, in a meta-analytical setting is a subject of future work.

The MOSTest method has several limitations. First, as with several other multivariate methods, it only provides a p-value, but does not provide the effect direction, limiting the application of several post-GWAS tools. Second, currently, MOSTest requires the availability of raw genotype data. Third, while discrete phenotypes will turn into continuous scales after pre-residualization and inverse-normal transformation, we have not formally validated whether MOSTest is applicable to case/control traits. Removing these limitations is also a subject of future work.

With the large gain of power and consequently lower required sample sizes of MOSTest, we predict that it will be possible to uncover the majority of SNPs influencing brain morphology in the upcoming years. The UKB initiative, for instance, is set to release neuroimaging data of a 100,000 individuals by 2022[25], which we estimate will enable MOSTest to identify SNPs that explain about 40% of the additive genetic variance in regional brain morphology. MOSTest is well-suited as an exploratory tool, followed up by studies that investigate the relation between the set

of discovered loci and individual features, with a much decreased multiple-comparisons burden. The MOSTest code is user friendly and publicly available, see the Supplementary Materials. In addition to brain structure, the MOSTest approach may also be of value in uncovering the genetic determinants of brain function and other complex human phenotypes consisting of correlated component measures, such as mental, cognitive, or cardiometabolic phenotypes, by taking advantage of the rich multivariate data sets now available.

## Methods

**Sample**. We made use of data from participants of the UKB population cohort, obtained from the data repository under accession number 27412. The composition, set-up, and data gathering protocols of the UKB have been extensively described elsewhere[26]. For this study, we selected individuals that had undergone the neuroimaging protocol and had White European ancestry, as determined by a combination of self-identification as "White British" and similar genetic ancestry based on genetic principal components. For the primary analysis, making use of T1 MRI scan data released up to April 2019, we excluded 1094 individuals with a primary or secondary ICD10 diagnosis of a neurological or mental disorder, as well as 594 individuals with bad structural scan quality as indicated by an age and sex-adjusted Euler number[27] more than three standard deviations lower than the scanner site mean. Our sample size for this analysis was $n = 26,502$, with a mean age of 55.51 years (SD = 7.42). 51.97% of the sample was female. For the replication analyses, we made use of an additional neuroimaging batch released in September 2019. After identical preprocessing steps as the primary sample, this sample consisted of 4884 individuals with a mean age of 55.47 years (SD = 7.37), 52.42% was female.

**Data preprocessing**. $T_1$-weighted scans were collected from three scanning sites throughout the United Kingdom, all on identically configured Siemens Skyra 3T scanners, with 32-channel receive head coils. The UKB core neuroimaging team has published extensive information on the applied scanning protocols and procedures, which we refer to for more details[25]. The $T_1$ scans were obtained from the UKB data repositories and stored locally at the secure computing cluster of the University of Oslo. We applied the standard "recon-all -all" processing pipeline of Freesurfer v5.3, performing automated surface-based morphometry and subcortical segmentation[28,29]. From the output, we extracted the sets of regional subcortical and cortical morphology measures, as well as estimated intracranial volume (eICV). Supplementary Table 1 contains all the regional morphology measures, per subset, included in the current study. For each of these, we included both the left and right hemisphere measure, if applicable.

We subsequently regressed out age, sex, scanner site, Euler number, and the first 20 genetic principal components from each measure. We further regressed out a global measure specific to each of the feature subsets: eICV for the subcortical volumes, mean thickness for the regional thickness measures, and total surface area for the regional surface area measures. This was done to ensure we are studying the genetic determinants of regional brain morphology rather than global effects. Following this, we applied rank-based inverse-normal transformation[30] to the residuals of each measure: $y'_i = \Phi^{-1}\left(\frac{r_i - c}{N - 2c + 1}\right)$, where $r_i$ is the ordinary rank of the measure for $i$th individual, $N$ gives the sample size, $\Phi^{-1}$ denotes the standard normal quantile, $y'_i$ is the value after transformation, and $c = 0.5$. This leads to normally distributed input into the univariate GWAS. See Supplementary Fig. 12 and the associated text for a more in-depth discussion of the importance of this normalization procedure.

**Univariate GWAS procedure**. We made use of the UKB v3 imputed data, which has undergone extensive quality control procedures as described by the UKB genetics team[31]. After converting the BGEN format to PLINK binary format, we additionally carried out standard quality check procedures, including filtering out individuals with more than 10% missingness, SNPs with more than 5% missingness, and SNPs failing the Hardy–Weinberg equilibrium test at $p = 1 \times 10^{-9}$. We further set a minor allele frequency threshold of 0.005, leaving 7,428,630 SNPs.

The univariate GWAS on each of the 171 pre-residualized and normalized regional brain morphology measures were carried out using the standard additive model of linear association between genotype vector, $\mathbf{g_j}$, and phenotype vector, $\mathbf{y}$. Statistical significance was assessed from Pearson's correlation coefficient $r_j = \text{corr}(\mathbf{y}, \mathbf{g_j})$, as implemented in MATLAB's corr function. This is equivalent to testing significance of the regression slope, $\hat{\beta}_j$, as both $\hat{\beta}_j$ and $r_j$ are assumed to be $t$-distributed and have the same $t$-value: $t_j = \beta_j/\text{se}\left(\hat{\beta}_j\right) = r_j/\text{se}(r_j) = r_j\sqrt{N-2}/\sqrt{1-r_j^2}$, and therefore the same $p$-value, equal to Student's $t$-cumulative distribution function (cdf) with $N-2$ degrees of freedom: $P_{\text{val},j} = 2\text{tcdf}\left(-\left|t_j\right|, N-2\right)$, where $N$ is the sample size[16]. Further, we validated that the above procedure produces the same results as the association test implemented in the commonly used PLINK's additive model, option plink --assoc.

Independent significant SNPs and genomic loci were identified in accordance with FUMA SNP2GENE definition[21]. First, we select a subset of SNPs that pass genome-wide significance threshold $5 \times 10^{-8}$ (calculated by min-P or MOSTest), and use PLINK to perform a clumping procedure at LD $r^2 = 0.6$, to identify the list of independent significant SNPs. Second, we clump the list of independent significant SNPs at LD $r^2 = 0.1$ threshold to identify lead SNPs. Third, we query the reference panel for all candidate SNPs in LD $r^2$ of 0.1 or higher with any lead SNPs. Further, for each lead SNP, it's corresponding genomic locus is defined as a contiguous region of the lead SNPs' chromosome, containing all candidate SNPs in $r^2 = 0.1$ or higher LD with the lead SNP. Finally, adjacent genomic loci are merged together if they are separated by <250 Kb. Allele LD correlations are computed from EUR population of the 1000 genomes Phase 3 data.

**MOSTest procedure**. Let $z_{ij}$ be the value of signed test statistic ($z$-score) calculated from the univariate association test between $j$th SNP and $i$th phenotype. Let $\mathbf{z_j} = (z_{1j}, \ldots, z_{Kj})$ be the vector of $z$-scores of $j$th SNP across $K$ phenotypes. Let $\mathbf{Z} = \{z_{ij}\}$ be the matrix of $z$-scores, with rows corresponding to SNPs, and columns corresponding to phenotypes. Further, let $\widetilde{\mathbf{Z}} = \left\{\widetilde{z}_{ij}\right\}$ be the matrix of $z$-scores, calculated from association tests on a randomly permuted genotype vector of each SNP. To preserve correlation structure among phenotypes, the permutation was performed only once for each SNP, and the resulting genotype vector was used in association test across all phenotypes.

The MOSTest test statistic, $X_j^2$, for the $j$th SNP is calculated as Mahalanobis norm $X_j^2 = \mathbf{z_j^T}\widetilde{\mathbf{R}}^{-1}\mathbf{z_j}$, where $\widetilde{\mathbf{R}}$ is the $K$-by-$K$ correlation matrix of $\widetilde{\mathbf{Z}}$. The null hypothesis of the MOSTest is that $z_j$ is distributed as a multivariate normal random variable with zero mean and covariance $\widetilde{\mathbf{R}}$. To compute the theoretical (i.e., under null) $p$-value of the MOSTest test statistic, we calculated the tail probability that a Chi-square statistic exceeds $X_j^2$. This probability is given by chi-square distribution with K degrees of freedom, or, equivalently, a gamma distribution, gamma(K/2,2)[32]. Instead of using theoretical values, we fit the two free parameters of the gamma($a$, $b$) distribution to the observed distribution of $X_j^2$ under permutation (shown in Supplementary Table S4). The $p$-value of the MOSTest test statistic is then obtained from a cumulative distribution function of the gamma distribution, $p_{\text{MOST}} = \text{CDF}_{\text{gamma}(a,b)}(\mathbf{z_j^T}\widetilde{\mathbf{R}}^{-1}\mathbf{z_j})$. To clarify our notation, gamma($a$, $b$) distribution is parametrized by shape ($a$) and scale ($b$), so that $p_{\text{gamma}}(x|a,b) = \frac{1}{b^a\Gamma(a)}x^{a-1}e^{-\frac{x}{b}}$, where $\Gamma(\cdot)$ is the gamma function.

In the simulation scenarios with regularization, the correlation matrix $\widetilde{\mathbf{R}}$ matrix was replaced with its regularized version $\widetilde{\mathbf{R}}' = \mathbf{US_r'U^T}$, where $\mathbf{USU^T} = \widetilde{\mathbf{R}}$ is an SVD decomposition of $\widetilde{\mathbf{R}}$, and the diagonal matrix $S_r'$ was obtained from S by replacing $r$ smallest eigenvalues with the next smallest eigenvalue; $r$ is an integer parameter that runs from 0 (no regularization) to K-1 (full regularization).

Controlling for covariates, such as genetic principal components, is done via pre-residualization of all phenotype vectors, i.e., we replace them with the corresponding residual after multiple linear regression of the phenotype vector on the covariates. In addition, we perform a rank-based inverse-normal transformation of the residualized phenotypes, to ensure that $z$-scores forming the input to MOSTest are normally distributed.

MOSTest code is publicly available: https://github.com/precimed/mostest.

**MiXeR analysis**. We applied a causal mixture model[6,16] to estimate the percentage of variance explained by genome-wide significant SNPs as a function of sample size. For each SNP, $i$, MiXeR models its additive genetic effect of allele substitution, $\beta_i$, as a point-normal mixture, $\beta_i = (1 - \pi_1)N(0, 0) + \pi_1 N(0, \sigma_\beta^2)$, where $\pi_1$ represents the proportion of non-null SNPs (polygenicity) and $\sigma_\beta^2$ represents variance of effect sizes of non-null SNPs (discoverability). Then, for each SNP, $j$, MiXeR incorporates LD information and allele frequencies for 9,997,231 SNPs extracted from 1000 Genomes Phase 3 data to estimate the expected probability distribution of the signed test statistic, $z_j = \delta_j + \epsilon_j = N \sum_i \sqrt{H_i}r_{ij}\beta_i + \epsilon_j$, where $N$ is sample size, $H_i$ indicates heterozygosity of $i$th SNP, $r_{ij}$ indicates allelic correlation between $i$th and $j$th SNPs, and $\epsilon_j \sim N(0, \sigma_0^2)$ is the residual variance. Further, the three parameters, $\pi_1, \sigma_\beta^2, \sigma_0^2$, are fitted by direct maximization of the likelihood function. Fitting the univariate MiXeR model does not depend on the sign of $z_j$, allowing us to calculate $|z_j|$ from MOSTest $p$-values, $\left|z_j\right| = |F^{-1}\left(p_j/2\right)|$, where $F^{-1}$ is the inverse function of the standard normal c.d.f., and $p_j$ is $p$-value from MOSTest. Finally, given the estimated parameters of the model, the power curve $S$ ($N$) is then calculated from the posterior distribution $p(\delta_j|z_j, N)$.

**Gene-set analyses**. We made use of the Functional Mapping and Annotation of GWAS (FUMA) online platform (https://fuma.ctglab.nl/) to further process the output from MOSTest and min-P. Through FUMA, we carried out MAGMA-based gene analyses using default settings, which entail the application of a SNP-wide mean model and use of the 1000 Genomes Phase 3 EUR reference panel. Gene-set analyses were done in a similar manner, restricting the sets under

investigation to those that are part of the Gene Ontology biological processes subset ($n = 4436$), as listed in the Molecular Signatures Database (MsigDB) v5.2.

**Simulations.** In our simulations, we used the same panel of $N = 26,502$ individuals and $M = 7,428,630$ markers as in our main analysis. Using a simple additive genetic model, we drew genetic effects $\beta$ from a certain distribution, and then calculated quantitative phenotypes $y_k$ of the $k$th sample as $y_k = \sum_{j=1}^{M} g_{kj}\beta_j + \epsilon_k$, where $\mathbf{G} = (g_{kj})$ is an $N$ by $M$ genotype matrix, containing the number of reference alleles for the $k$th sample and $j$th variant. Here, $\beta_j$ is the causal effect size, and $\epsilon$ is a normally distributed residual, drawn from a normal distribution with zero mean and variance set in a way that yields the pre-defined level of heritability, $h^2 = \mathrm{Var}(\mathbf{G}\beta)/\mathrm{Var}(\mathbf{y})$. The computations were performed using publicly available tools (https://github.com/precimed/simu).

To generate the vector of genetic effects, $\beta$, we randomly drew a certain number (typically nc = 100) of "causal" variants from chromosome 21 (containing 102,079 SNPs). The genetic effect sizes of all other variants were set to zero. Under the assumption of negligible linkage disequilibrium across chromosomes, this design allowed us to validate a uniform $p$-value distribution under null (i.e., on all chromosomes except chromosome 21), thus confirming a correct type-I error. In addition, we validated the type-I error under the genotype permutation scheme employed by the MOSTest procedure. Across causal variants, the $\beta$ values were drawn either from a normal distribution, or from a standard Cauchy distribution. The default settings, particularly nc = 100 and h2 = 0.004, were chosen to simulate a realistic magnitude of the genetic effects, with h2 = 29.1% heritability and nc = 7.3 K causal variants genome-wide, if all chromosomes would have been allowed to have genetic effects.

To simulate a realistic multivariate scenario with genetic and phenotypic correlation across up to $T = 25$ features, we introduced covariance structure both in $\beta$ and in $\epsilon$, independently set either to an identity matrix (no correlation) or to the correlation matrix of the first $T$ subcortical volumes, as defined in our main analysis. Further, for each causal genetic variant, we vary the number of non-zero effects across features, $t$, ranging from $t = 1$ (sparse effects scenario), to $t = T$ (distributed effects scenario). In the sparse cases, $t < T$, we increased the total number of causal variants proportionally to the ratio of $T/t$, and ensured that each trait had exactly the pre-defined number of causal variants, while each causal variant, on average, contributed to $t$ out of $T$ traits.

We applied the link function to generate non-normally distributed phenotypes with heavy tails, yielding $\mathbf{y} = f(\mathbf{G}\beta + \epsilon)$, where $f$ can be either identity, $f(x) = x$, or an exponent $f(x) = e^x$. To simulate strict linear dependency between features (i.e., a rank-deficient covariance matrix), we replaced the set of $T$ features, calculated as $\mathbf{y} = \mathbf{G}\beta + \epsilon$, with a set of $T(T - 1)/2$ features, formed by all pairwise combinations. We then applied the link function $f$ (identity or an exponent).

A detailed list of simulation parameters and scenarios is specified in Supplementary Tables 4 and 5. Each simulation scenario was performed ten times, each with random selection of causal variants, and randomly drawn $\beta$ and $\epsilon$.

Finally, we executed MQFAM, MultiPhen and MultiABEL according to guidelines provided by the original software. A typical simulation result of a single run visualizes type-I error under null, type-I error under permutations, statistical power to detect non-null association at causal variants, and corresponding quantile–quantile plots (on chromosome 21, under null, and under permutations). Full simulation scripts are available at https://github.com/precimed/mostest

**Reporting summary**. Further information on research design is available in the Nature Research Reporting Summary linked to this article.

## Data availability

The data incorporated in this work were gathered from the public UK Biobank resource. The generated GWAS summary statistics can be fetched from https://archive.sigma2.no/pages/public/datasetDetail.jsf?id=10.11582/2020.00031.

## Code availability

The code is available via https://github.com/precimed/mostest (GPLv3 license).

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

## Acknowledgements

We were funded by the Research Council of Norway (276082, 213837, 223273, 204966/F20, 229129, 249795/F20, 225989, 248778, 249795), the South-Eastern Norway Regional Health Authority (2013-123, 2014-097, 2015-073, 2016-064, 2017-004), Stiftelsen

Kristian Gerhard Jebsen (SKGJ-Med-008), The European Research Council (ERC) under the European Union's Horizon 2020 research and innovation programme (ERC Starting Grant, Grant Agreement No. 802998) and National Institutes of Health (R01MH100351, R01GM104400, NIDA/NCI: U24DA041123). This work was partly performed on the TSD (Tjeneste for Sensitive Data) facilities, owned by the University of Oslo, operated and developed by the TSD service group at the University of Oslo, IT-Department (USIT). (tsd-drift@usit.uio.no). Computations were also performed on resources provided by UNINETT Sigma2—the National Infrastructure for High Performance Computing and Data Storage in Norway. This work used the Extreme Science and Engineering Discovery Environment (XSEDE) including COMET and OASYS resources at the UCSD through allocation TG-IBN200001.

## Author contributions

A.M.D., O.F., D.v.d.M., and O.A.A. conceived the study; D.v.d.M., O.F., T.K., and A.M.D. pre-processed the data. D.v.d.M., O.F., and A.M.D. performed all analyses, with conceptual input from O.A.A.; A.A.S., A.D., O.B.S., W.K.T., C.C.F., D.H., and L.T.W. contributed to interpretation of results; D.v.d.M. and O.F. drafted the manuscript and all authors contributed to and approved the final manuscript.

## Competing interests

Dr. Andreassen has received speaker's honorarium from Lundbeck, and is a consultant to HealthLytix. Dr. Dale is a Founder of and holds equity in CorTechs Labs, Inc, and serves on its Scientific Advisory Board. He is a member of the Scientific Advisory Board of Human Longevity, Inc. and receives funding through research agreements with General Electric Healthcare and Medtronic, Inc. The terms of these arrangements have been reviewed and approved by UCSD in accordance with its conflict of interest policies. The other authors declare no competing interests.
