## [Peer Review File · Nature Communications]

Reviewers' comments:

Reviewer #1 (Remarks to the Author):

Making the MOSTest of imaging genetics
Van der Meer et al.

This paper proposed a multivariate omnibus statistical test, termed MOSTest, for joint analysis of brain morphological measurements in a multivariate framework. The authors showed that compared with existing methods such as the min-P approach, MOSTest has substantially increased power for discovery. The authors applied MOSTest to the UK Biobank imaging genetic dataset and identified many new genetic loci associated with imaging measurements relative to the recent ENIGMA study which has a much larger sample size.

Overall I found that this paper is well-motivated and clearly written. However, I have some major concerns as detailed below.

* The simulation study in this work is way too simplified. This section should be greatly expanded and corresponding methods and results should be presented in the main text. In particular, it is critical to ensure that the proposed method does not have inflated type I error under a wider range of situations. The authors only conducted one set of simulations that perfectly follows the assumptions made by MOSTest. But what happens if the effects of variants are not shared across all traits but only a subset of the traits? What happens if the distribution of SNP effect sizes are not Gaussian? What happens if phenotypes have different heritability? All these + other potential factors that may affect the validity of the test should be systematically investigated under the simulation framework.

* When evaluating the control of type I error, it's not enough to report the false positive rate at the nominal 0.05; although the parametric tail approximation looks good on Figure S2, more formal assessment of the behavior of extreme tails and the accuracy of the approximation is warranted.

* Relevant to the points above, as there is no free lunch in statistics, one statistical test is unlikely to be the most powerful in all situations. In this particular case, although it's totally reasonable to assume that many genetic variants have distributed effects across morphological measurements, I can imagine that the power of MOSTest will decrease when variants have more specific (less distributed) effects. The authors should compare the power of MOSTest with alternative methods, build power curves, and discuss when the multivariate test may not work well. Empirically, the authors demonstrated that MOSTest identified many more SNPs than the ENIGMA study, but is there any SNP that was identified by univariate tests but missed by MOSTest? A more detailed comparison between univariate and multivariate methods + some discussions along this line will be helpful.

* Based on the description, \tilde{R} was computed across all SNPs. Is there any concern that estimation of this correlation matrix is biased by LD patterns? I.e., SNPs in long LD blocks contribute more to the correlation? Would that be better, both for accuracy and computational cost, if only a set of independent SNPs is used for the calculation of \tilde{R} ?

* The author should double check if they really used v3 imputed data. The total number of v3 SNPs should be much larger than 7.4M.

* This preprint (<https://www.biorxiv.org/content/biorxiv/early/2019/09/05/758326.full.pdf>) developed a highly similar statistical test to the MOSTest presented here, and was posted to

bioRxiv before MOSTest. The authors should give them more credits. In fact, there should be a section in introduction that reviews existing multi-trait association tests, rather than giving an incomplete literature review in the extended data.

* Regarding the Extended Data section, in addition to the simulation and literature review which should be moved and expanded, I think it'd be helpful to presenting more on the replication results in the main text as well.

Reviewer #2 (Remarks to the Author):

van der Meer et al. implemented and applied a pipeline named MOSTest for multivariate analysis of GWAS data on imaging phenotypes, and discovered more loci associated with sets of imaging traits. Although I personally like seeing more multivariate GWAS and more discoveries, the paper is not written in a justified manner. The authors over-sell the pipeline as a new method without careful investigations while at the same time do not go into more interesting details of their discoveries. There appears to be a lot of work remain to make the story a compelling case. My comments are as follows.

Major points:

1. The biggest issue of the authors' presentation is that they focus on selling MOSTest as a new method/pipeline that boosts discovery power via a multivariate test. I have to say that there is not sufficient novelty in the pipeline that makes MOSTest a novel method. For example, lines 66-72, 1) applying rank-based transformation to the residualized phenotypes in a standard protocol; 2) estimating phenotypic correlation from GWAS summary statistics is not new at all, e.g. Stephens 2013 (<https://journals.plos.org/plosone/article?id=10.1371/journal.pone.0065245>) and Zhu et al. 2015 ([https://www.cell.com/ajhg/fulltext/S0002-9297\(14\)00477-7](https://www.cell.com/ajhg/fulltext/S0002-9297(14)00477-7)); 3) the so-called Mahalanobis norm appears to be an over-fancy name for the classic Hotelling's T-squared MANOVA test statistic, and MANOVA is used in multivariate GWAS, e.g. Stephens 2013 (<https://journals.plos.org/plosone/article?id=10.1371/journal.pone.0065245>) and Shen et al. 2017 (<https://www.nature.com/articles/s41467-017-00453-3>), and even what's implemented in MV-PLINK is a MANOVA test statistic; 4) the Gamma density function fitting part is redundant (see point 3 below).
2. The majority of the comparisons were carried out versus the min-P approach. This is not a fair comparison that justifies MOSTest. min-P is obviously underpowered, and the power gain of multivariate methods against univariate analysis has been studied a lot (e.g. Porter & O'Reilly 2017 <https://www.nature.com/articles/srep38837>). When the authors compare to MV-PLINK, it is obvious as I said above, these are all MANOVA thus perform very similarly. So I can't see the point comparing to min-P. What should have been investigated are: What kind of additional signals are detected by the multivariate test? How many are in e.g. Grasby et al.'s 255 loci? What do these additional associations mean in terms of e.g. genotype-phenotype map?
3. The authors emphasize that they conduct a permutation-based test for the multivariate GWAS. This is not true. The permutation implemented in the paper is to obtain a reasonable estimate of the R matrix (which is more or less just the phenotypic correlation matrix, e.g. Zhu et al. 2015 [https://www.cell.com/ajhg/fulltext/S0002-9297\(14\)00477-7](https://www.cell.com/ajhg/fulltext/S0002-9297(14)00477-7)), but not the empirical distribution of the genome-wide p-values under the null, thus not a permutation-based test for the GWAS. Thereafter, the authors fit a Gamma distribution for the test statistic under the null, which is to me redundant. As I said above, the test statistic is standard MANOVA Hotelling's T-squared, which simply follows a K-df chi-squared distribution. This is also clear according to Supp Table 4.
4. Line 84-85, the authors believe that the power of the multivariate test comes from the presence of pleiotropy. This is related to the interpretation of multivariate results. I'm almost certain that all

such MANOVA tests mainly gain power from phenotypic correlations rather than pleiotropy-induced genetic correlations. When two traits are highly correlated, even if the SNP is only affecting one trait, the bivariate test will have better power than the univariate test. Thus, an essential question to ask and answer is: which traits are driving the multivariate signal?

5. Line 115-116, 40% replication rate for both multivariate and univariate methods at a 5% threshold. Does this mean an FDR of 12.5%? That seems high. The authors should think about a better way of reporting the replication analysis.

6. Line 121-123, the authors claim that MOSTest runs 10,000 times faster than MV-PLINK. This is again not a fair comparison as the former is a summary-statistics-based pipeline whereas the latter is an individual-level-data-based tool. If the authors really want to justify their speed, the pipeline should be compared to existing multivariate GWAS tools that also use summary statistics as input, e.g. MTAG (<https://www.nature.com/articles/s41588-017-0009-4>; software: <https://github.com/omeed-maghzian/mtag>) and MultiABEL (<https://www.nature.com/articles/s41467-017-00453-3>; software: <https://github.com/xiashen/MultiABEL>)

7. Line 203-205, the authors claim that the output for MOSTest is suited for secondary analyses and follow-up investigations. This bothers me a lot. If I understand correctly, the authors simply transform the MANOVA p-value into a 1-df chi-squared value and use it as a single genetic effect estimate "beta" for the multi-trait test. This doesn't make practical sense. First of all, such a genetic effect has no direction; Second, it doesn't reflect the source trait(s) of the multivariate association; Third, it doesn't tell us anything about the genotype-phenotype map (see also point 2 above); Fourth, the distribution of this "beta" is bounded and highly skewed (irregular), therefore the standard error derived from p-value is not a proper one, e.g. no good for Wald test, etc. Most GWAS follow-up analysis tools do use proper genetic effect estimates together with their standard errors, but it doesn't mean they can take these "beta" and "se". For example, 1) how can the authors be sure that these estimates are proper for the MiXeR tool which fits a Gaussian mixture? 2) are these estimates compatible with LD structure for e.g. GCTA-COJO conditional analysis and LD score regression? I doubt it - the correlation between such betas can hardly be consistent with the LD correlations of the markers; 3) are these estimates suitable for an MR analysis for causal inference? I will not list all possible analyses we can do using GWAS summary statistics, but they cannot simply take any estimates as "betas".

8. LD score regression intercept was used to validate genomic inflation. First of all, this should be interpreted with care. Evidence is available showing that the intercept is not independent of population structure, e.g. Yengo et al. (2018) <https://www.biorxiv.org/content/10.1101/310565v1>. Second, see point 7 above, I don't even think the authors' multivariate betas can be justified to use tools like LD score regression.

9. Fig. 4, do the genes in the analyzed GO sets overlap? If they do, these sets are hardly independent. The authors should clarify the overlapping extent in order not to misinterpret the results. Related to this, are there particular loci/genes driving these enrichments?

Other points (not necessarily minor):

10. The software is not available to the referees. Also, I can't see any problem having the tool on GitHub before the paper gets published.

11. Fig. 2 y-axis, estimated percent "genetic" variance.

12. Line 379-381, subscripts are messed. The authors say $Z = \{z_{ij}\}$ have SNPs as the rows and phenotypes as the columns, but $z_j = (z_{1j}, \dots, z_{kj})$ are for the j-th SNP across K phenotypes?

13. Row 390 should be with K degrees of freedom instead of N.

Reviewer #3 (Remarks to the Author):

I would like to thank the authors for this interesting article, very well written and concise. I think that the subject is timely, the processing and computing effort are also impressive. I believe that the manuscript could be of great interest for the community and would benefit from a more comprehensive comparison with current methods: MV-PLINK [already partially discussed] and MTAG (Turley et al., 2018).

1) For the comparison with MV-PLINK, I am surprised by the computing time you report. Is this using PLINK 1.9 or more recent? In my experience (but I may be wrong), MV in plink should not take that long and doing k univariate GWAS often takes longer than running a single k multivariate one. I think some data is showed in See Porter and Reilly, 2017 (STable 3).

1.1) If computation is possible, I think it should be done genome wide to really compare the number of associations. The good agreement presented in FigS1, selects ~300 significant SNPs but we do not know if more significant results could be found using MV-PLINK. It could also be tested in the simulations to compare its FWER and power to that of MOSTest.

1.2) In addition, I think it would help the reader to position your method in regards to MTAG that models genetic correlation and sample overlap to combine GWAS summary statistics. If meaningful, it would be great to compare the results obtained via the 2 methods.

2) I think it would be good to briefly discuss the fact that what you may gain in power you somewhat lose in precision, the SNPs you find being globally associated with the brain measurements. In this regard, I find Fig3 would benefit for reporting the pvalues or mentioning which associations with cortical ROIs are significant.

3) How would you interpret the matrix R? If I understand well, it is the correlation between GWAS effect sizes of the different phenotypes, after permuting once each SNP. What is the difference compared to using a matrix of genetic correlation estimated from LDscore regression (or GCTA) for example? Would it be equivalent to what you are doing? I think a discussion would help the readers.

Minor:

4) "While cortical thickness and surface area have been reported to be phenotypically and genetically only weakly correlated to each other⁷, many brain-related traits, e.g. mental disorders, share a large proportion of genetic variants, even in the absence of an overall correlation⁸"
Not sure what you mean with the "absence of overall correlation", the comorbidity of mental health disorders suggests a correlation.

5) "For this study, we selected White Europeans that had undergone the neuroimaging protocol"
Did you define this using genetic PCs?

6) "minor allele frequency threshold of .005, leaving 7.4 million SNPs"
This may include a lot of SNPs with little variation (and no homozygotes), could you report the MAF of the top SNPs in the supplementary tables?

7) "the min-P approach"

I thought this was called TATES. Where does the min-P name come from, and is it used in the ENIGMA paper?

8) "This probability is given by chi-square distribution with N degrees of freedom"

I think that you mean K, not N?

Is it $\text{Gamma}(K/2, 0.5)$? or $\text{Gamma}(K/2, 2)$ (as per Stable 4) ?

8) Did you do any visual QC as per the ENIGMA guidelines? Some of the ROI boundaries can be unstable and biased for some participants.

10) Would MOSTest work on discrete phenotypes? Rank transformation would not necessarily be meaningful here.

11) For your simulations on odd-even chromosome, how many replicates did you perform? What is the estimated type 1 error of the test? And its power?

Also am I correct to say that without rank transformation, MOSTest has greatly reduced power (Figure S3)? Could you also report type 1 error and power in that case?

12) Which language your program is implemented in?

We would like to thank the editor for a chance to revise the manuscript, and the reviewers for their insightful comments. We believe we have addressed their concerns, as outlined below, and that this has strengthened the manuscript. In accordance with the reviewer comments, we have significantly expanded the set of simulations to validate the behavior of MOSTest under a wider range of conditions, and we have included additional comparisons with other multivariate methods. We also now describe the MOSTest in terms that more fully consider the existing multivariate GWAS literature. Please find the responses to each individual reviewer comment below.

Reviewer #1 (Remarks to the Author):

This paper proposed a multivariate omnibus statistical test, termed MOSTest, for joint analysis of brain morphological measurements in a multivariate framework. The authors showed that compared with existing methods such as the min-P approach, MOSTest has substantially increased power for discovery. The authors applied MOSTest to the UK Biobank imaging genetic dataset and identified many new genetic loci associated with imaging measurements relative to the recent ENIGMA study which has a much larger sample size.

Overall, I found that this paper is well-motivated and clearly written. However, I have some major concerns as detailed below.

1). The simulation study in this work is way too simplified. This section should be greatly expanded and corresponding methods and results should be presented in the main text. In particular, it is critical to ensure that the proposed method does not have inflated type I error under a wider range of situations. The authors only conducted one set of simulations that perfectly follows the assumptions made by MOSTest. But what happens if the effects of variants are not shared across all traits but only a subset of the traits? What happens if the distribution of SNP effect sizes are not Gaussian? What happens if phenotypes have different heritability? All these + other potential factors that may affect the validity of the test should be systematically investigated under the simulation framework.

Response

We thank the Reviewer for this comment. We fully agree that this study benefits from an expanded set of simulations, investigating the validity of MOSTest and other multivariate tests under a wider range of situations. Generally, we have used a framework whereby we simulate causal effects on chromosome 21, containing 102 thousand SNPs, while we keep the other chromosomes free of genetic signal in order to validate correct type 1 error. Specifically, in line with the above questions, we have carried out the following simulations:

- a) **Sparse vs. distributed effects:** we have calculated the performance of multivariate tests following simulation of shared non-zero effects on 1 (sparse), 4 or 10 (distributed) features, while varying the total number of included features from 1 to 100.
- b) **Distribution of causal effect sizes:** we have simulated a Cauchy distribution (as a heavy-tailed distribution), in addition to a normal distribution.
- c) **Vary heritability (h^2):** we have simulated $h^2=0.004$, 0.04 , and 0.4 for chromosome 21. We have further simulated an extreme scenario where a subset of traits were not heritable ($h^2=0$), which represents the most challenging scenario for the multivariate methods.
- d) **Other scenarios:** we have carried out a range of simulations to further probe the influence of polygenicity, the importance of the inverse normal transformation, presence vs absence of phenotypic and genetic effects correlations, varying complexity of the feature space,

including non-normally distributed features and a degenerative covariance matrix due to linear dependency across features, and the use of the phenotypic correlation matrix R instead of permutation-based correlation matrix \tilde{R} .

We prioritize the above simulation scenarios over other alternatives covering more complex distributions of SNPs across causal variants (such as, for example, LD- and MAF- dependent genetic architectures, of differential enrichment across genomic annotations). The rationale for omitting these simulations is that, in the MOSTest procedure, the two steps that involve calculations across multiple SNPs (namely, estimation of the correlation matrix \tilde{R} , and fitting the gamma function) are done under genotype permutation scheme, which makes the distribution of genetic effects irrelevant. All other MOSTest calculations are performed on non-permuted genotypes, but those are specific to a given SNP, thus the distribution across SNPs is also not relevant.

The simulations are now described in the Results and Online Methods (see also below). To summarize:

Under a range of conditions, all multivariate methods showed similar statistical power, while min-P had consistently lower power (Figure S1). The methods also all maintained correct type-I equally well under the null and following permutation, under these conditions.

Interestingly, the min-P approach outperformed the multivariate methods under one specific condition: when a small set of heritable features are analyzed together with a much larger set of non-heritable features (Figure S3).

The simulations demonstrated the importance of the rank-based inverse-normal transformation: without this transformation, the tests had inflated type-I error, as well as lower statistical power, when the features were not normally distributed (Figure S5).

Our simulations further show that spectral regularization, implemented only in MOSTest, is essential when there is linear dependence between features (Figure S6).

Our extensive validation of the MOSTest methodology and implementation show that the MOSTest is a valid statistical test with good statistical power to detect associations in a multivariate context, across a wide range of conditions.

Lines 167-210: “We performed extensive validation of MOSTest methodology and implementation, checking its performance under a range of conditions through simulations with synthetic data, and comparing this with other multivariate approaches besides min-P, namely MultiABEL, MultiPhen, and MQFAM. We used a framework whereby effects are simulated on chromosome 21 to compute statistical power, while all other chromosomes are kept free of genetic signal to estimate type-I error under the null. Tables S4 and S5 list the full set of simulation scenarios, such as varying the sparsity of genetic effects and the number of features included, as well as the heritability of these features and their correlation structure. Further details about the simulation framework are provided in the Online Methods, with additional data in the Extended Data section.

Under a range of conditions, all multivariate methods showed similar statistical power, except for min-P with lower power. The methods also all maintained correct type-I equally well under the null and following permutation, under these conditions, as summarized in Figures S1 and S2. We could however not explicitly validate type-I error for MultiPhen and MQFAM across the genome due to slow runtime; these typically exceeded one hour to calculate p-values per $M=100$ variants in the

power analyses, for each one of 760 simulation runs. Therefore, it was impractical to run these tests for M=7.3 million causal variants. See Table S5 for the runtimes per simulation.

The min-P approach outperformed the multivariate methods under one specific condition: when a small set of heritable features are analyzed together with a much larger set of non-heritable features, see Figure S3. All tests have similar power when all features are heritable, but do not share genetic variants, or when shared genetic effect sizes follow a heavy-tailed distribution, see Figure S4.

The simulations revealed the importance of the rank-based inverse-normal transformation: without this transformation, the tests had inflated type-I error, as well as lower statistical power, when the features were not normally distributed, see Figure S5. We note that an incorrect type I error fully prohibits an application of a statistical test. The MOSTest genotype permutation scheme, absent in other tools, effectively captures this issue, therefore forms an important part of the test that prevents reporting inflated p-values and spurious associations. Further, Figure S5 shows the role of covariance structure across genetic effects, and across features. We observe that the highest power to detect associations occurs when features have realistic covariance structure, but genetic effects are not correlated.

Our simulations further show that spectral regularization, implemented only in MOSTest, is essential when there is linear dependence between features. Regularization was not necessary in our main analysis, as the conditioning number of phenotypic correlation matrix R was reasonably low (Table S4), leading to a well-defined matrix R-1. However, in the presence of linear dependence, MOSTest has invalid type-I error without spectral regularization, while correct regularization solves this issue and at the same time improves statistical power (Figure S6). Further, estimation of phenotype covariance matrix can be done either from the phenotypes themselves, or from z-scores under permutation. The schemes yield nearly identical results, which also holds in all other simulation scenarios (data not shown). Note that the permutation scheme has the advantage of not requiring availability of all phenotypes at one site, allowing for application in a meta-analytical setting.

Please see Online Methods and Supplementary Material for further validation data, including Figure S7, displaying scaled-down version of the simulations, and Figure S8, QQ plots showing that the MiXeR model correctly captures the LD dependence of the MOSTest association statistics.”

Lines 426-465: “In our simulations, we use the same panel of N=26,502 individuals and M=7,428,630 markers as in our main analysis. Using a simple additive genetic model, we draw genetic effects β from a certain distribution, and then calculate quantitative phenotypes y_k of k -th sample as $y_k = \sum_{j=1}^M g_{kj} \beta_j + \epsilon_k$, where $G = (g_{kj})$ is N by M genotype matrix, containing the number of reference alleles for k -th sample and j -th variant, β_j is the causal effect size, and ϵ is a normally distributed residual, drawn from a normal distribution with zero mean and variance set in a way that yields pre-defined level of heritability, $h^2 = \text{var}(G\beta)/\text{var}(y)$. The computations were performed using publicly available tools (<https://github.com/precimed/simu>).

To generate the vector of genetic effects, β , we randomly draw a certain number (typically $nc=100$) of “causal” variants from chr21 (containing $M_{21}=102,079$ SNPs), setting all genetic effect sizes of all other variants to zero. Under the assumption of negligible linkage disequilibrium across chromosomes, this design allows us to validate a uniform p-value distribution under null (i.e. on all chromosomes but chr21), thus confirming a correct type-I error. Additionally, we validate the type-I error under the genotype permutation scheme employed by the MOSTest procedure. Across causal variants, the β values are drawn either from a normal distribution, or from a standard Cauchy distribution. The default settings, particularly $nc=100$ and $h^2=.004$, are chosen in a way that yields

realistic magnitude of the genetic effects, corresponding to $h^2=29.1\%$ heritability and $n_c=7.3K$ causal variants genome-wide, if all chromosomes would have been allowed to have genetic effects.

To simulate a realistic multivariate scenario with genetic and phenotypic correlation across up to $T=25$ features, we introduce covariance structure both in β and in ϵ , which can be independently set either to an identity matrix (no correlation), or to the correlation matrix of the first T subcortical volumes, as defined in our main analysis. Further, for each causal genetic variant we vary the number of non-zero effects across features, t , ranging from $t = 1$ (sparse effects scenario), to $t = T$ (distributed effects scenario). In the sparse cases, $t < T$, we increase the total number of causal variants proportionally to the ratio of T/t , and ensure that each trait has exactly the pre-defined number of causal variants, while each causal variant, on average, contributes to t out of T traits.

To generate non-normally distributed phenotypes with heavy tails we apply the link function, yielding $y = f(G\beta + \epsilon)$, where f can be either identity, $f(x) = x$, or an exponent $f(x) = e^x$. To simulate strict linear dependency between features (i.e. a rank-deficient covariance matrix), we replace the set of T features, calculated as $y = G\beta + \epsilon$, with a set of $T(T - 1)/2$ features, formed by all pairwise combinations, and afterwards apply the link function f (identity or an exponent).

A detailed list of simulation parameters and scenarios is specified in Supplementary Table S4 and S5. Each simulation scenario is performed 10 times, each with random selection of causal variants, and randomly drawn β and ϵ .

Finally, we execute MQFAM, MultiPhen and MultiABEL according to guidelines provided by the original software. A typical simulation result of a single run visualizes type-I error under null, type-I error under permutations, statistical power to detect non-null association at causal variants, and corresponding quantile-quantile plots (on chr21, under null, and under permutations). Full simulation scripts are available at <https://github.com/precimed/mostest>.

2). When evaluating the control of type I error, it's not enough to report the false positive rate at the nominal 0.05; although the parametric tail approximation looks good on Figure S2, more formal assessment of the behavior of extreme tails and the accuracy of the approximation is warranted.

Response

We have now greatly expanded our simulation framework, as reported above, and with it also our reporting of the type-I error under all these conditions. For each simulation, we list the type-I error across five different p-value thresholds (0.05, 0.01, 1e-4, 1e-6 and 5e-8), under the null (no effects simulated on all chromosomes, except chr21), with error bars indicating standard deviations across 10 runs. Please see Supplementary Figures S1-S7 for the results.

3). Relevant to the points above, as there is no free lunch in statistics, one statistical test is unlikely to be the most powerful in all situations. In this particular case, although it's totally reasonable to assume that many genetic variants have distributed effects across morphological measurements, I can imagine that the power of MOSTest will decrease when variants have more specific (less distributed) effects. The authors should compare the power of MOSTest with alternative methods, build power curves, and discuss when the multivariate test may not work well. Empirically, the authors demonstrated that MOSTest identified many more SNPs than the ENIGMA study, but is there any SNP that was identified by univariate tests but missed by MOSTest? A more detailed comparison between univariate and multivariate methods + some discussions along this line will be helpful.

Response

The ability of MOSTest to pick up on distributed effects, and the gain this provides over alternative methods lies at the core of this study. We therefore fully agree that a greater exploration of the presence of sparse versus distributed effects, as well as the performance under different situations, is in order. We have now significantly expanded our simulations to examine the behavior and power of MOSTest vs other approaches under a range of conditions. This includes, as mentioned under point 1), simulations of sparse to distributed effects. In accordance with the reviewer, we have included plots showing power to detect effects in each simulation, for both MOSTest and other approaches. We found overall that MOSTest and other multivariate approaches performed well under a wide range of conditions compared to min-P. We did find one situation where min-P outperformed the multivariate approaches, indeed in the presence of sparse effects, when there were a low number of heritable features with shared genetic effects among many non-heritable features. This result is now described in the main manuscript, and shown in Figure S3:

Lines 183-185: “The min-P approach outperformed the multivariate methods under one specific condition: when a small set of heritable features are analyzed together with a much larger set of non-heritable features, see Figure S3.”

Lines 273-280: “Our simulations showed that in some scenarios min-P outperforms the multivariate methods, specifically when genetic signal is sparse across phenotypes. As the level of sparsity will be different across SNPs, it is expected that min-P may discover a few additional SNPs that are missed by MOSTest, as also indicated in our comparison on real data. We are planning to develop an automated regularization strategy, to select the best possible trade-off between univariate and current multivariate methods, to further improve power of the multivariate analysis.”

Further, as requested, we provide more information on the loci identified by MOSTest versus min-P: we have listed all lead SNPs identified through either min-P only (20), through MOSTest only (255), or through both (92), together with how many and which regions these SNPs had a whole-genome significant effect on in Data S1-S3:

Lines 116-121: “For all features combined, 92 loci were discovered by both MOSTest and min-P, 20 were unique to min-P, and 255 were only discovered by MOSTest. Data S1-S3 lists these loci, together with for how many and which regions the lead SNPs were significant. Generally, loci discovered by both tests had significant effects on multiple regions, loci discovered only by min-P had effects on 1 or 2 regions, and loci discovered only by MOSTest often had no whole-genome significant effect on any of the regions.”

4). Based on the description, \tilde{R} was computed across all SNPs. Is there any concern that estimation of this correlation matrix is biased by LD patterns? I.e., SNPs in long LD blocks contribute more to the correlation? Would that be better, both for accuracy and computational cost, if only a set of independent SNPs is used for the calculation of \tilde{R} ?

Response

Yes, \tilde{R} is computed across all SNPs. However, this is done under permutation, breaking LD structure. Note that, the correlation structure across phenotypes is preserved as the permutation is carried out only once, using the resulting permuted genotype vector for the association test across all phenotypes. We have now carried out a simulation whereby we use the phenotypic correlation matrix instead of \tilde{R} , showing the schemes yield nearly identical results. Please see Figure S6. Note

that the permutation scheme has the advantage of not requiring availability of all phenotypes at one site, allowing for application in a meta-analytical setting. This is now described as follows:

Lines 203-207: “Further, estimation of phenotype covariance matrix can be done either from the phenotypes themselves, or from z-scores under permutation. The schemes yield nearly identical results, which also holds in all other simulation scenarios (data not shown). Note that the permutation scheme has the advantage of not requiring availability of all phenotypes at one site, allowing for application in a meta-analytical setting.”

5). The author should double check if they really used v3 imputed data. The total number of v3 SNPs should be much larger than 7.4M.

Response

Indeed, the number of SNPs is lower than in several other UKB genetics studies. We can however ensure that we have used the v3 imputed data; the difference is due to the QC filters we apply, (93M variants in total, with correct imputation for non-HRC variants). The difference is due to the QC filters we apply, primarily the use of a minor allele frequency (MAF) filter of .005 in a sample of 26502 subjects with imaging data.

6). This preprint (<https://www.biorxiv.org/content/biorxiv/early/2019/09/05/758326.full.pdf>) developed a highly similar statistical test to the MOSTest presented here, and was posted to bioRxiv before MOSTest. The authors should give them more credits. In fact, there should be a section in introduction that reviews existing multi-trait association tests, rather than giving an incomplete literature review in the extended data.

Response

Thank you for this suggestion. In addition to expanding our comparisons, we have now included such a review of relevant multivariate GWAS approaches in the introduction. We thank the Reviewer for pointing to the aMAT preprint, which was posted nearly simultaneously with our preprint. We have now described aMAT in the introduction. The central point of aMAT is the regularization of the phenotypic matrix, which we did not use in our main analysis, because of an invertible \tilde{R} matrix. Regularization may, indeed, provide a further improvement in power of the multivariate analysis. We have mentioned this as a future direction for MOSTest development in the Discussion.

Lines 62-84: “Several multivariate approaches to GWAS have been proposed to date^{3,11}. In this context, *multivariate association* at a given single nucleotide polymorphism (SNP) means that at least one of multiple traits being considered is associated with the genotype vector of that SNP¹¹. To test for multivariate association, MQFAM (also known as MV-PLINK)⁵ and MultiPhen⁴ both perform a multiple regression whereby the genotype vector is used as an outcome variable, while each phenotype is turned into an explanatory variable; the p-value is then calculated from an F-test, which tests for an association between the genotype vector and the most predictive linear combination of phenotypes at each SNP. The advantage of MultiPhen is that it uses ordinal regression, while MQFAM is based on canonical correlation analysis, which in theory is less appropriate for prediction of a categorical variable such as 0-1-2 coded genotype vector. The statistical power of both methods is known to be similar to MANOVA³. MultiABEL is another multivariate GWAS approach, which implements Pillai’s trace MANOVA¹² to calculate multivariate p-value from summary statistics; its authors further advocate the importance of rank-based inverse normal transformation. Most

recently, aMAT¹³ was introduced. This test, based on a chi-square test statistic, explores regularization (spectral filtering) of the correlation matrix R as a way to further boost statistical power in multivariate methods. Distinct from these multivariate tests of association is the Multi-Trait Analysis of GWAS (MTAG) method¹⁴, which boosts discovery in a primary trait by conditioning on one or more secondary traits and therefore does not test the multivariate null hypothesis that none of the traits are associated with a given SNP. While all multivariate methods listed above have been shown to substantially increase gene discovery compared to univariate approaches³, they have not been widely adopted by large-scale international consortia. This may result from too high computational costs, lack of user friendliness, lack of method validation, and/or model assumptions not fitting with the real data.“

Lines 277-280: “We are planning to develop an automated regularization strategy, in line with the aMAT method¹², to select the best possible trade-off between univariate and current multivariate methods, to further improve power of the multivariate analysis.”

7). Regarding the Extended Data section, in addition to the simulation and literature review which should be moved and expanded, I think it'd be helpful to presenting more on the replication results in the main text as well.

Response

Thank you, we agree. As mentioned above, we have now included this literature review in the introduction, and placed much of the expanded simulation framework in the main manuscript (lines 167-210 and 426-465, see other answers). We have moved Table S3, now Table 1, listing the replication results, to the main text as well. Further, note that we now list the replication p-values, both through MOSTest and min-P, per discovered locus in Data S1-S3.

Reviewer #2 (Remarks to the Author):

van der Meer et al. implemented and applied a pipeline named MOSTest for multivariate analysis of GWAS data on imaging phenotypes, and discovered more loci associated with sets of imaging traits. Although I personally like seeing more multivariate GWAS and more discoveries, the paper is not written in a justified manner. The authors over-sell the pipeline as a new method without careful investigations while at the same time do not go into more interesting details of their discoveries. There appears to be a lot of work remain to make the story a compelling case. My comments are as follows.

Major points:

1). The biggest issue of the authors' presentation is that they focus on selling MOSTest as a new method/pipeline that boosts discovery power via a multivariate test. I have to say that there is not sufficient novelty in the pipeline that makes MOSTest a novel method. For example, lines 66-72, 1) applying rank-based transformation to the residualized phenotypes is a standard protocol; 2) estimating phenotypic correlation from GWAS summary statistics is not new at all, e.g. Stephens 2013 (<https://journals.plos.org/plosone/article?id=10.1371/journal.pone.0065245>) and Zhu et al. 2015 ([https://www.cell.com/ajhg/fulltext/S0002-9297\(14\)00477-7](https://www.cell.com/ajhg/fulltext/S0002-9297(14)00477-7)); 3) the so-called Mahalanobis norm appears to be an over-fancy name for the classic Hotelling's T-squared MANOVA test statistic, and MANOVA is used in multivariate GWAS, e.g. Stephens 2013 (<https://journals.plos.org/plosone/article?id=10.1371/journal.pone.0065245>) and Shen et al. 2017 (<https://www.nature.com/articles/s41467-017-00453-3>), and even what's implemented in MV-PLINK is a MANOVA test statistic; 4) the Gamma density function fitting part is redundant (see point 3 below).

Response

We agree and thank the reviewer for these references. We have now included these throughout the manuscript, and have added MultiABEL to our new set of comparisons of multivariate methods.

We agree that we should not make any claims that the individual steps of the method are novel, and agree that the text should have been clearer on this topic. We have now rephrased our wording throughout the manuscript, removing all claims that the MOSTest is a novel statistical method. Rather than novelty, the intended central message of this manuscript is the distributed nature of genetic signal across neuroimaging measures, and how the combination of the mentioned steps, comprising MOSTest, is important to optimize the power to detect this signal. Thus, we use the multivariate method to test the hypothesis that SNPs associated with brain morphology phenotypes are distributed across brain regions and measures. The large increase in discovery certainly supports this hypothesis. We have now revised the text to make this clearer.

To further support the combination of steps making up MOSTest, we have greatly expanded our set of simulations to show the power and type-I error rate of MOSTest and the mentioned existing multivariate methods. Please see the revised text in the Results and Online Methods sections, also listed in the answer to Reviewer #1 comment #1. We found that, under a range of conditions, all multivariate methods showed similar statistical power and maintained correct type-I equally well under the null. However, they also revealed some important advantages of MOSTest, now summarized in the Discussion as follows (please see the new Results and Online Methods section for the details):

“Key advantages of MOSTest include (1) orders of magnitude faster run-time as compared to MQFAM and MultiPhen, (2) a built-in genotype permutation scheme that allows detection of cases of invalid type-I error, (3) the ability to incorporate spectral regularization of the phenotype matrix, unavailable in MultiABEL, and (4) extensive validation through simulations.”

The rationale for fitting the gamma function is to enable regularization of the covariance matrix. We elaborate on this in the answer for point 3 below.

We would further like to mention that an advantage of MOSTest, compared to several other multivariate GWAS methods, e.g. MANOVA, is that it works in two stages: first we compute univariate summary statistics, and then we combine them into a multivariate p-value. Note that the second stage does not require access to raw genotypes. MOSTest can therefore, in principle, be applied in a meta-analytical setting, after combining univariate results across sites with e.g. standard variance-based meta-analysis. This is a subject of future work, as now mentioned in the Discussion:

Line 282-283: “Enabling the MOSTest procedure, and particularly its genotype permutation scheme, in a meta-analytical setting is also a subject of future work.”

2) The majority of the comparisons were carried out versus the min-P approach. This is not a fair comparison that justifies MOSTest. min-P is obviously underpowered, and the power gain of multivariate methods against univariate analysis has been studied a lot (e.g. Porter & O'Reilly 2017 <https://www.nature.com/articles/srep38837>). When the authors compare to MV-PLINK, it is obvious as I said above, these are all MANOVA thus perform very similarly. So I can't see the point comparing to min-P. What should have been investigated are: What kind of additional signals are detected by the multivariate test? How many are in e.g. Grasby et al.'s 255 loci? What do these additional associations mean in terms of e.g. genotype-phenotype map?

Response

The key message of this study is the gain in yield that can be obtained when considering the distributed nature of genetic signal across correlated component measures, tested on the brain morphology use case (imaging genetics). This is why the comparison with the widely used min-P was chosen, including the largest imaging genetics studies to date (e.g. Grasby *et al.* 2020), that clearly does not consider this shared signal across measures. In other words, the primary aim of this manuscript is to test the hypothesis that a method designed to capture shared signal is superior for imaging genetics, not to provide an exhaustive comparison of multivariate GWAS methods.

With that key message in mind, we agree it is important to provide more information about the difference between MOSTest and min-P in detected signals, and more generally look into the detection of sparse versus distributed effects. We have therefore carried out analyses of the behavior of MOSTest when simulating sparse to distributed effects. We found that, as expected, min-P has lower power under a wide range of conditions, but it did outperform the multivariate methods under one specific condition: when a small set of heritable features are analyzed together with a much larger set of non-heritable features, please see the Results section and Figure S3.

We agree that it is of interest to clarify what types of signals are detected by MOSTest versus min-P, and how these map onto the phenotypes. Therefore, we have now created new tables, for all features combined, that list the loci that are discovered by both tests (92 loci), those identified by MOSTest alone (255) and those identified by min-P only (20), as now described in the first paragraph

of the Results section. These tables list for the lead SNP of each of these loci how many regions it had a significant effect on, and which regions.

Lines 116-121: " for all features combined, 92 loci were discovered by both MOSTest and min-P, 20 were unique to min-P, and 255 were only discovered by MOSTest. Data S1-S3 lists these loci, together with for how many and which regions the lead SNPs were significant. Generally, loci discovered by both tests had significant effects on multiple regions, loci discovered only by min-P had effects on 1 or 2 regions, and loci discovered only by MOSTest often had no whole-genome significant effect on any of the regions."

3). The authors emphasize that they conduct a permutation-based test for the multivariate GWAS. This is not true. The permutation implemented in the paper is to obtain a reasonable estimate of the R matrix (which is more or less just the phenotypic correlation matrix, e.g. Zhu et al. 2015 [https://www.cell.com/ajhg/fulltext/S0002-9297\(14\)00477-7](https://www.cell.com/ajhg/fulltext/S0002-9297(14)00477-7)), but not the empirical distribution of the genome-wide p-values under the null, thus not a permutation-based test for the GWAS. Thereafter, the authors fit a Gamma distribution for the test statistic under the null, which is to me redundant. As I said above, the test statistic is standard MANOVA Hotelling's T-squared, which simply follows a K-df chi-squared distribution. This is also clear according to Supp Table 4.

Response

We apologize for the unclear wording. As the reviewer points out, our p-values are not based on a permutation test. We have therefore now removed any claim that MOSTest is permutation-based. At the same time, we show in the revised manuscript the importance of the genotype permutation scheme, as it provides means to detect invalid type-I error, which would prohibit application of a test, or indicate the need to regularize the covariance matrix. In terms of user experience, turning this into a built-in feature is an important advantage of MOSTest over MultiABEL.

Regarding the fitting of a gamma distribution, the results in the main text indeed imply this is redundant. However, in certain cases, it is necessary to regularize the R matrix, e.g. using spectral regularization. Fitting a Gamma distribution is important in these cases, as depending on regularization strategy the resulting test statistic may not follow the chi square distribution. We have now included simulations showing that spectral regularization, implemented only in MOSTest, is essential when there is linear dependence between features. Without spectral regularization, MOSTest has invalid type-I error, while correct regularization solves this issue, and at the same time improves statistical power (Figure S6).

Lines 197-202: "Our simulations further show that spectral regularization, implemented only in MOSTest, is essential when there is linear dependence between features. Regularization was not necessary in our main analysis, as the conditioning number of phenotypic correlation matrix R was reasonably low (Table S4), leading to a well-defined matrix R^{-1} . However, in the presence of linear dependence, MOSTest has invalid type-I error without spectral regularization, while correct regularization solves this issue and at the same time improves statistical power (Figure S6)."

4). Line 84-85, the authors believe that the power of the multivariate test comes from the presence of pleiotropy. This is related to the interpretation of multivariate results. I'm almost certain that all such MANOVA tests mainly gain power from phenotypic correlations rather than pleiotropy-induced genetic correlations. When two traits are highly correlated, even if the SNP is only affecting one trait,

the bivariate test will have better power than the univariate test. Thus, an essential question to ask and answer is: which traits are driving the multivariate signal?

Response

The statistical power of the multivariate tests considered in this paper depends both on the correlation structure across traits, and across genetic effects. We now show this through simulations with either realistic ('real') or no ('eye') covariance structure across the genetic effects ('rg') and/or the phenotype residuals ('re'; Figure S5). Importantly, the multivariate tests show an advantage over minP also when both phenotypes and genotypes are uncorrelated. Therefore, we believe it is incorrect to state that boost in power comes from either phenotypic or genetic correlation. However we observed that the highest statistical power is observed when there is a correlation among phenotypes, and absent correlation among genetic effects. To explain this behavior, we provide an illustrative example below, showing a bivariate normal distribution with strong correlation (representing the expected distribution of Z scores under null), and two SNPs – a SNP "A" that follows the correlation pattern, and a SNP "B" that show an opposite pattern of effect direction. Both SNPs would have equal p-value in multivariate analysis, however SNP "B" has lower z-scores (in magnitude).

5). Line 115-116, 40% replication rate for both multivariate and univariate methods at a 5% threshold. Does this mean an FDR of 12.5%? That seems high. The authors should think about a better way of reporting the replication analysis.

Response

Our replication sample size is relatively small ($n=4,884$), contributing to a relatively low replication rate. However, the replication rate was consistent between min-P and MOSTest, indicating that the validity of SNPs implicated in multivariate analysis is as strong as those discovered by min-P. We have now added the replication table to the main manuscript, and also listed the replication p-values, generated by MOSTest and min-P, per discovered locus in Data S1-S3.

6). Line 121-123, the authors claim that MOSTest runs 10,000 times faster than MV-PLINK. This is again not a fair comparison as the former is a summary-statistics-based pipeline whereas the latter is an individual-level-data-based tool. If the authors really want to justify their speed, the pipeline

should be compared to existing multivariate GWAS tools that also use summary statistics as input, e.g. [MTAG \(https://www.nature.com/articles/s41588-017-0009-4\)](https://www.nature.com/articles/s41588-017-0009-4); software: <https://github.com/omeed-maghzian/mtag> and [MultiABEL \(https://www.nature.com/articles/s41467-017-00453-3\)](https://www.nature.com/articles/s41467-017-00453-3); software: <https://github.com/xiashen/MultiABEL>

Response

Please note that we did include include the runtime of the univariate GWAS (both original and imputed) for MOSTest in this comparison. We therefore do believe the runtime is a fair criterion to point to in our comparison with MV-PLINK (MQFAM). We have now further included comparisons with other multivariate tools, namely MultiABEL and MultiPhen. While MultiPhen suffers from the same problem as MQFAM of a practically infeasible runtime with the number of included features, variants and individuals, MultiABEL did run nearly as fast as MOSTest (not counting the runtime of the univariate GWAS). We have now included an overview of runtimes in Table S6.

We did not include a comparison with MTAG, as this has a rather different underlying concept: For a given SNP, MTAG provides a p-value for a primary trait, and it only shows a boost if this trait correlates phenotypically with other traits in the datasets. MOSTest and other multivariate GWAS approaches aim to answer whether the SNP is associated with at least one (or more) of the traits. We now include a paragraph introducing existing multivariate GWAS methods in the Introduction (lines 62-84, revised text also listed under reviewer #1 comment #6) and describe the comparisons extensively in the Results section (Lines 167-210, revised text also listed under reviewer #1 comment #1).

7). Line 203-205, the authors claim that the output for MOSTest is suited for secondary analyses and follow-up investigations. This bothers me a lot. If I understand correctly, the authors simply transform the MANOVA p-value into a 1-df chi-squared value and use it as a single genetic effect estimate "beta" for the multi-trait test. This doesn't make practical sense. First of all, such a genetic effect has no direction; Second, it doesn't reflect the source trait(s) of the multivariate association; Third, it doesn't tell us anything about the genotype-phenotype map (see also point 2 above); Fourth, the distribution of this "beta" is bounded and highly skewed (irregular), therefore the standard error derived from p-value is not a proper one, e.g. no good for Wald test, etc. Most GWAS follow-up analysis tools do use proper genetic effect estimates together with their standard errors, but it doesn't mean they can take these "beta" and "se". For example, 1) how can the authors be sure that these estimates are proper for the MiXeR tool which fits a Gaussian mixture? 2) are these estimates compatible with LD structure for e.g. GCTA-COJO conditional analysis and LD score regression? I doubt it - the correlation between such betas can hardly be consistent with the LD correlations of the markers; 3) are these estimates suitable for an MR analysis for causal inference? I will not list all possible analyses we can do using GWAS summary statistics, but they cannot simply take any estimates as "betas".

Response

We apologize for the imprecise wording, leading to misinterpretation. The original sentence read "The output of MOSTest is suited for secondary analyses and follow-up studies to investigate the relation between the set of loci discovered and individual features, with a much decreased multiple-comparisons burden." We did not intend for this to be interpreted as a statement that the MOSTest summary statistics broadly lend themselves for further analysis by post-GWAS biostatistical tools. Rather, we meant that the great statistical power of MOSTest makes it an excellent tool for GWAS discovery, implicating a subset of SNPs. The effects of this subset of SNPs, on e.g. the univariate

features, can then be investigated further, in follow-up studies, with lower multiple comparisons burden. We acknowledge that the sentence needs to be revised, and we have therefore replaced this sentence with the following:

Lines 293-296: "MOSTest is well-suited as an exploratory tool, followed up by studies that investigate the relation between the set of discovered loci and individual features, with a much decreased multiple-comparisons burden."

Regarding the validity of MOSTest for the MiXeR tool, we have now included the accompanying QQ plots, showing that our model fits the data well, see Figure S8. We have further clarified the transformation from MOSTest p-values to z-scores, used as input to the MiXeR model ($|z_j| = |F^{-1}(p_j/2)|$, where F^{-1} is the inverse function of the standard normal c.d.f., and p_j is p-value from MOSTest)

Line 412-416. "Fitting the univariate MiXeR model does not depend on the sign of z_j , allowing us to calculate $|z_j|$ from MOSTest p-values, $|z_j| = |F^{-1}(p_j/2)|$, where F^{-1} is the inverse function of the standard normal c.d.f., and p_j is p-value from MOSTest."

We acknowledge that the lack of a direction of effect impedes the use of MOSTest summary statistics for several post-GWAS analysis tools. We have now mentioned this in the discussion as a limitation of MOSTest:

Lines 284-286: "First, as with several other multivariate methods, it only provides a p-value, but does not provide the effect direction, limiting the application of some post-GWAS tools."

8). LD score regression intercept was used to validate genomic inflation. First of all, this should be interpreted with care. Evidence is available showing that the intercept is not independent of population structure, e.g. Yengo et al. (2018) <https://www.biorxiv.org/content/10.1101/310565v1>. Second, see point 7 above, I don't even think the authors' multivariate betas can be justified to use tools like LD score regression.

Response

We fully agree with the reviewer's concern. We have now revised the manuscript and explicitly mentioned that the LD score regression results should be interpreted with care, and that we have not provided formal proof that MOSTest scales with LD structure.

Lines 212-214: "Note that the LD score regression results should be interpreted with care¹⁷, and that we have not provided formal proof that MOSTest p-values scales with LD structure as required by LD score regression model."

9). Fig. 4, do the genes in the analyzed GO sets overlap? If they do, these sets are hardly independent. The authors should clarify the overlapping extent in order not to misinterpret the results. Related to this, are there particular loci/genes driving these enrichments?

Response

Yes, the gene sets identified as significant through competitive gene-set analyses, following multiple comparison corrections, overlap. There were 4436 Biological Processes Gene Ontology sets in our analyses, each containing between about a 100 and 2000, out of a total of 19142, protein-coding genes, there is therefore substantial overlap between all of them. In order to clarify the overlap between the reported sets, we have now added a matrix depicting the overlap (ratio to total genes in the pathway in the upper triangle, absolute number of genes overlapping in the lower triangle) between each pair to the supplementary material. We thank the reviewer for this suggestion. We further now include Table S7, with an ordered list of all multiple comparisons-corrected significant pathways, and Table S8, containing all significant genes ranked by their p-values, listing to which of these gene sets they belong.

Lines 238-240 "Please see Figure S9 for the overlap between these pathways, and the Extended Data for further information on all significant genes and genetic pathways."

Other points (not necessarily minor):

10). The software is not available to the referees. Also, I can't see any problem having the tool on GitHub before the paper gets published.

Response

We apologize for our software not being available. We did provide a .zip package with software, including example data and a tutorial, upon initial submission. We will be in contact with the editorial staff to ensure this will be available to the reviewers upon resubmission. Further, we have now made it public on Github <https://github.com/precimed/mostest>

11). Fig. 2 y-axis, estimated percent "genetic" variance.

Response

Thank you for pointing this out. We have now adjusted this, as shown in the new Figure 2.

12). Line 379-381, subscripts are messed. The authors say $Z=\{z_{ij}\}$ have SNPs as the rows and phenotypes as the columns, but $z_j=(z_{1j}, \dots, z_{kj})$ are for the j-th SNP across K phenotypes?

Response

Thank you for pointing this out. We have now adjusted this:

Lines 370-371: ("Let $Z=\{z_{ij}\}$ be the matrix of z-scores, with columns corresponding to SNPs, and rows corresponding to phenotypes")

13). Row 390 should be with K degrees of freedom instead of N.

Response

Thank you for pointing this out. We have now adjusted this.

Reviewer #3 (Remarks to the Author):

I would like to thank the authors for this interesting article, very well written and concise. I think that the subject is timely, the processing and computing effort are also impressive. I believe that the manuscript could be of great interest for the community and would benefit from a more comprehensive comparison with current methods: MV-PLINK [already partially discussed] and MTAG (Turley et al., 2018).

1) For the comparison with MV-PLINK, I am surprised by the computing time you report. Is this using PLINK 1.9 or more recent? In my experience (but I may be wrong), MV in plink should not take that long and doing k univariate GWAS often takes longer than running a single k multivariate one. I think some data is showed in See Porter and Reilly, 2017 (STable 3).

Response

Thank you for pointing out this table. Our data differs widely from what is used in Porter and O'Reilly's study with regards to the number of traits included, as well as the number of genetic variants, and individuals, which may explain the difference. We have now greatly expanded our simulation framework, including comparisons with other multivariate methods (see other answers), and included the runtime for each simulation in Table S5. This further confirms that MOSTest is fast, with this type of data, compared to the other methods.

Note that MQFAM functionality is absent in official versions of the PLINK, including plink 1.07, plink 1.9 and plink 2.0. Instead, we used a separate executable available here:

<https://genepi.qimr.edu.au/staff/manuelF/multivariate/usage.html>

1.1) If computation is possible, I think it should be done genome wide to really compare the number of associations. The good agreement presented in FigS1, selects ~300 significant SNPs but we do not know if more significant results could be found using MV-PLINK. It could also be tested in the simulations to compare its FWER and power to that of MOSTest. 1.2) In addition, I think it would help the reader to position your method in regards to MTAG that models genetic correlation and sample overlap to combine GWAS summary statistics. If meaningful, it would be great to compare the results obtained via the 2 methods.

Response

Unfortunately, the computational cost of 250,000 hours is too high to allow for running this genome-wide. We did significantly expand our comparisons with other multivariate methods, namely with multiABEL and multiPhen. Further, the simulations show very similar power between all multivariate methods, with nearly overlapping QQ plots. Therefore, we believe further comparison of MQFAM and MOSTest on real data is not warranted. We did not include a comparison with MTAG, as this has a rather different underlying concept: For a given SNP, MTAG provides a p-value for a primary trait, and it only shows a boost if this trait correlates phenotypically with other traits in the datasets. MOSTest and other multivariate GWAS approaches do not make such a distinction between primary and secondary traits, they aim to answer whether the SNP is associated with at least one (or more) of the traits. We now describe multivariate GWAS methods in the introduction, and the comparisons of these methods with MOSTest are described in detail in the Results section:

2). I think it would be good to briefly discuss the fact that what you may gain in power you somewhat lose in precision, the SNPs you find being globally associated with the brain measurements. In this regard, I find Fig3 would benefit for reporting the pvalues or mentioning which associations with cortical ROIs are significant.

Response

Thank you for this suggestion. We agree that more information on the different types of signal that are identified by MOSTest compared to univariate methods/min-P is important, as it is closely tied to our core message of the distributed nature of genetic signal across brain measures and how MOSTest can pick up such signal. We have therefore now included information on the number of loci identified by both MOSTest and min-P (92 loci), by MOSTest only (255), and by min-P only (20). Further, we now list for the lead SNP of each of these loci on how many, and which, regions these had a significant effects. Please see Data S1-S3.

We have further carried out analyses of the behavior of the contrasted methods when simulating sparse to distributed effects. These added analyses are described in the Results section, please see also supplementary Figure S3. In general, we find that MOSTest and other multivariate methods have better power than min-P under a wide range of conditions, but that min-P does outperform them in the specific scenario where a small set of heritable features are analyzed together with a much larger set of non-heritable features. This is now also mentioned in the Discussion section:

Lines 273-280: “Our simulations showed that in some scenarios min-P outperforms the multivariate methods, specifically when genetic signal is sparse across phenotypes. As the level of sparsity will be different across SNPs, it is expected that min-P may discover a few additional SNPs that are missed by MOSTest, as also indicated in our comparison using real data. We are planning to develop an automated regularization strategy, to select the best possible trade-off between univariate and current multivariate methods, to further improve power of the multivariate analysis.”

Note that we now also describe the explorative nature of MOSTest in the Discussion, that it may be followed up by studies of the individual features:

Lines 293-296: “MOSTest is well-suited as an exploratory tool, followed up by studies that investigate the relation between the set of discovered loci and individual features, with a much decreased multiple-comparisons burden.”

Last, with regard to Figure 3 in the main manuscript, we chose to report Z-values instead of P-values to convey direction of effects, note that an absolute Z-value of 5.45 corresponds to whole-genome significance ($p=5*10^{-8}$), we now include this statement in the figure legend.

Line 230-232: “Note: the absolute Z-value scaling is clipped at 8 ($p=1.2*10^{-15}$); an absolute Z-value of 5.45 corresponds to whole-genome significance ($p=5*10^{-8}$).”

3). How would you interpret the matrix R? If I understand well, it is the correlation between GWAS effect sizes of the different phenotypes, after permuting once each SNP. What is the difference compared to using a matrix of genetic correlation estimated from LDscore regression (or GCTA) for example? Would it be equivalent to what you are doing? I think a discussion would help the readers.

Response

Matrix \tilde{R} (estimated from z-score correlation under MOSTest genotype permutation scheme) is, conceptually, a way of estimating the phenotypic correlation matrix, R. As we have both matrices available to use, we have now shown in simulations that using \tilde{R} vs R yields nearly identical results, please see Figure S6. The reason to prefer \tilde{R} is that in the future, expanding MOSTest to meta-analytical setting, we will not have the phenotypic correlation matrix R across sub-studies, while combining Z scores will still be possible. We have now described this as follows:

Lines 203-207: “Further, estimation of phenotype covariance matrix can be done either from the phenotypes themselves, or from z-scores under permutation. The schemes yield nearly identical results, which also holds in all other simulation scenarios (data not shown). Note that the permutation scheme has the advantage of not requiring availability of all phenotypes at one site, allowing for application in a meta-analytical setting.”

Minor:

4). *“While cortical thickness and surface area have been reported to be phenotypically and genetically only weakly correlated to each other⁷, many brain-related traits, e.g. mental disorders, share a large proportion of genetic variants, even in the absence of an overall correlation⁸”* Not sure what you mean with the “absence of overall correlation”, the comorbidity of mental health disorders suggests a correlation.

Response

Apologies for the lack of clarity here. With “overall correlation” in the last section of the sentence we refer to genetic correlations. We have now added the word “genetic” here. Indeed, the presence of comorbidity alludes to our point that these disorders may have large genetic overlap despite known low genetic correlations.

Lines 51-54: “While cortical thickness and surface area have been reported to be phenotypically and genetically only weakly correlated to each other, many brain-related traits, e.g. mental disorders, share a large proportion of genetic variants, even in the absence of an overall genetic correlation.”

5). *“For this study, we selected White Europeans that had undergone the neuroimaging protocol”* Did you define this using genetic PCs?

Response:

This was based on UKB data field 22006, which follows from a combination of self-identification as 'White British' and similar genetic ancestry based on genetic PCs. We now clarify this in the Online Methods section:

Lines 305-308: “For this study, we selected individuals that had undergone the neuroimaging protocol and had White European ancestry, as determined by a combination of self-identification as 'White British' and similar genetic ancestry based on genetic principal components.”

6) *“minor allele frequency threshold of .005, leaving 7.4 million SNPs”*

This may include a lot of SNPs with little variation (and no homozygotes), could you report the MAF of the top SNPs in the supplementary tables?

Response:

Thank you for this suggestion. Yes, we have now included this information in the SNP tables (Data S4-11)

7). *“the min-P approach” I thought this was called TATES. Where does the min-P name come from, and is it used in the ENIGMA paper?*

Response:

TATES is a variant of the min-P approach, both are based on the minimal p-value across the set of traits. Whereas the standard min-P approach adjusts these univariate P-values through Šidák correction, using the Nyholt estimation of the effective number of tests, the TATES approach makes an adjustment based on the P-value correlation matrix as determined by the phenotypic correlation matrix. ENIGMA used this TATES variant.

8). *“This probability is given by chi-square distribution with N degrees of freedom”*

I think that you mean K, not N? Is it Gamma(K/2,0.5)? or Gamma(K/2,2) (as per Stable 4) ?

Response

Thank you for pointing this out, it should indeed be K degrees of freedom. We have now adjusted this. For the second argument of the Gamma probability distribution, we have used inconsistent parametrization: b=0.5 in the Gamma(K/2, 0.5) formula referred to the “rate” parameter, while b=2.0 in Table S4 referred to the “scale” parameter. We have replaced gamma(K/2, 0.5) with gamma(K/2, 2), thus using “scale” parametrization consistently throughout the manuscript. We also provided an explicit formula for the probability density function and clarify the ambiguity regarding scale and rate parametrization of the Gamma function.

Lines 379-386: “This probability is given by chi-square distribution with K degrees of freedom, or, equivalently, a gamma distribution, $\text{gamma}(K/2,2)^{32}$. Instead of using theoretical values, we fit the two free parameters of the gamma(a,b) distribution to the observed distribution of X_j^2 under permutation (shown in Table S4). The p-value of the MOSTest test statistic is then obtained from a cumulative distribution function of the gamma distribution, $p_{MOST} = CDF_{\text{gamma}(a,b)}(z_j^T \hat{R}^{-1} z_j)$. To clarify our notation, gamma(a,b) distribution is parametrised by shape (a) and scale (b), so that $p_{\text{gamma}}(x|a,b) = \frac{1}{b^a \Gamma(a)} x^{a-1} e^{-\frac{x}{b}}$, where $\Gamma(\cdot)$ is the gamma function.”

9). *Did you do any visual QC as per the ENIGMA guidelines? Some of the ROI boundaries can be unstable and biased for some participants.*

Response

No, we did not do visual QC of the neuroimaging data. While we agree this could provide some benefit, the sheer size of the dataset makes this practically infeasible. Conservatively estimated, this would require more than a year of full-time painstaking work, with the judgements by the observers still being rather subjective.

Note that we did exclude all subjects whose scan had an Euler number, residualized for age and sex, more than three standard deviations below the scanner mean. Further, we pre-residualized the outcome measures for Euler numbers. Euler numbers are known to be correlated to head motion (Rosen *et al.* 2018), and we have previously shown that they also correlate with visual ratings of scan quality in cohort data (Kaufmann *et al.* 2019, Supplementary Figure 13).

Rosen, A. F. G. et al. Quantitative assessment of structural image quality. *Neuroimage* 169, 407–418 (2018).

Kaufmann, T. et al. Common brain disorders are associated with heritable patterns of apparent aging of the brain. *Nat. Neurosci.* 22, 1617–1623 (2019).

10). Would MOSTest work on discrete phenotypes? Rank transformation would not necessarily be meaningful here.

Response

We apply rank-based INT after covarying out globals and other covariates. Therefore, even a discrete phenotypes will turn into a fairly continuous scale. However, we have not formally validated that MOSTest is applicable to case/control traits. We now mention this in limitations and future work paragraphs of the Discussion section.

Lines 286-288: “Second, while discrete phenotypes will turn into continuous scales after pre-residualization and inverse-normal transformation, we have not formally validated whether MOSTest is applicable to case/control traits. This is also a subject of future work.”

11). For your simulations on odd-even chromosome, how many replicates did you perform? What is the estimated type 1 error of the test? And its power? Also am I correct to say that without rank transformation, MOSTest has greatly reduced power (Figure S3)? Could you also report type 1 error and power in that case?

Response

We have now greatly expanded our simulation framework, and with it the way we report type-I error and power. In the new simulations, we perform 10 replications per run, thus evaluating 73 million statistical test in each of the 76 simulation scenarios. Please see supplementary Figures S1-S7, for each of these bar plots.

Yes, it is correct that MOSTest has reduced power when non-normal features are not transformed. We have simulated features with several distributions and show the type-I error and power with and without transformations in Figure S5. This is now described in the main manuscript as follows:

Lines 188-190: “The simulations revealed the importance of the rank-based inverse-normal transformation: without this transformation, the tests had inflated type-I error, as well as lower statistical power, when the features were not normally distributed, see Figure S5.”

12). Which language your program is implemented in?

Response

MOSTest is currently implemented in MATLAB, and is now freely accessible at Github:
<https://github.com/precimed/mostest>

REVIEWERS' COMMENTS:

Reviewer #1 (Remarks to the Author):

The authors have addressed most of my concerns.

Lines 381-309 contain repetitive text.

The new text in the "Simulations" subsection is a bit hard to follow and doesn't provide the same level of clarity as other sections of the paper. I suggest the authors polish this section to improve clarity.

Reviewer #2 (Remarks to the Author):

The authors addressed some of my concerns adequately, but there are still questions remain unanswered. Although I appreciate the authors's efforts and willingness to focus on discoveries results instead of method selling, I would like to emphasize again that the key modules in the workflow are not novel in genetics. I summarize the specific concerns as follows.

1. Row 83: I don't agree with the stated reasons why multivariate analysis is not widely used in GWAS. The main reason is that the results are less interpretable and specific. This is actually why some of my questions were unanswered properly (see point 3-4 below).

2. In the response, the authors wrote:

"We would further like to mention that an advantage of MOSTest, compared to several other multivariate GWAS methods, e.g. MANOVA, is that it works in two stages: first we compute univariate summary statistics, and then we combine them into a multivariate p-value. Note that the second stage does not require access to raw genotypes. MOSTest can therefore, in principle, be applied in a meta-analytical setting, after combining univariate results across sites with e.g. standard variance-based meta-analysis".

This is not an advantage over other similar methods at all. MANOVA has already been implemented by different people/tools to analyze summary statistics directly, and it has already been applied to GWAS meta-analysis results. I can't see any advantage or novelty as claimed in this statement.

3. I asked what the additional associations from multivariate analysis mean in terms of genotype-phenotype map. I don't think this is answered. Let me be more clear, e.g. given the genotypes of the discovered SNPs, can we say anything more about what the phenotypes would look like?

4. I asked an essential question: which traits are driving the multivariate signal? Not answered. Surely the SNPs are detected to be associated with something in the whole group of imaging phenotypes, but which ones? Namely, we don't only want to know that a locus is found for imaging phenotypes, what's going on in the association? The multivariate replication also gives only a p-value, what's being replicated? How do we know the traits driving the signal in discovery samples are the same as those in replication?

5. The authors responded:

"Importantly, the multivariate tests show an advantage over minP also when both phenotypes and genotypes are uncorrelated. Therefore, we believe it is incorrect to state that boost in power comes from either phenotypic or genetic correlation."

What does "uncorrelated genotypes" mean? That doesn't sound like genetic correlation. If the two traits are independent, and the SNP affects only one trait, the multivariate test can hardly be more powerful than univariate minP even with Bonferroni correction, as MANOVA simply reduces to Fisher's method. Simple example: two independent traits, significance threshold 0.05, for one trait $p_1 = 0.02 < 0.05/2$ (significant), for the other trait $p_2 = 0.9$ (non-sig.), then Fisher's method multivariate $p = 0.09 > 0.05$ (non-sig.). General comment here, related to questions above: Do not over-sell multivariate analysis.

6. Figure 2: It seems MiXeR is used to estimate the percentage of variance explained by SNPs for a single trait. But in results related to Figure 2, the z-score was transformed from MOSTest which is based on multi-traits. Why can the "transformed z-score" from multi-trait be used for estimating the percentage of explained variance explained for a single trait?

Reviewer #3 (Remarks to the Author):

I would like to thank the authors for the clarifications and further analyses/simulations they performed, which makes a very compelling case in favour of using MOSTest for multivariate GWAS using univariate summary statistics. I would like to reiterate that this is a very well written manuscript, and that the MOSTest method could greatly improve SNP discovery. I only have a one minor comment (see below).

Lines 51-54: "While cortical thickness and surface area have been reported to be phenotypically and genetically only weakly correlated to each other, many brain-related traits, e.g. mental disorders, share a large proportion of genetic variants, even in the absence of an overall genetic correlation."

To me this still reads as if in there is not genetic correlation between mental disorders, which is not correct. See for example Antilla et al., 2018 (Brainstorm paper), who demonstrated what I would consider to be widespread genetic correlations between mental health diagnoses. Some of those rG are far from being small/trivial and one must keep in mind that SE may still be large.

Below, we outline our responses to the remaining concerns of the reviewers.

REVIEWERS' COMMENTS:

Reviewer #1 (Remarks to the Author):

The authors have addressed most of my concerns.

Lines 381-309 contain repetitive text.

Thank you for pointing this out. We have deleted the duplicated sentence:

“Instead of using theoretical values, we fit the two free parameters of the gamma(a, b) distribution to the observed distribution of X_j^2 under permutation (shown in Table S4). The p-value of the MOSTest test statistic is then obtained from a cumulative distribution function of the gamma distribution, $p_{\text{MOST}} = CDF_{\text{gamma}(a,b)}(\mathbf{z}_j^T \tilde{\mathbf{R}}^{-1} \mathbf{z}_j)$. “

The new text in the "Simulations" subsection is a bit hard to follow and doesn't provide the same level of clarity as other sections of the paper. I suggest the authors polish this section to improve clarity.

In accordance with the reviewer's suggestion, we have gone over this section and rewritten it for clarity.

Reviewer #2 (Remarks to the Author):

The authors addressed some of my concerns adequately, but there are still questions remain unanswered. Although I appreciate the authors's efforts and willingness to focus on discoveries results instead of method selling, I would like to emphasize again that the key modules in the workflow are not novel in genetics. I summarize the specific concerns as follows.

1. Row 83: I don't agree with the stated reasons why multivariate analysis is not widely used in GWAS. The main reason is that the results are less interpretable and specific. This is actually why some of my questions were unanswered properly (see point 3-4 below).

We have now rephrased that sentence in the introduction as follows:

“This may be due to the fact that the results can be less straightforward to interpret, in addition to high computational costs, lack of user friendliness, lack of method validation, and/or model assumptions not fitting with the real data.”

Note that we state the results can be less straightforward to interpret rather than less specific and interpretable *per se*. In a statistical sense, we control for specificity (i.e. true negative rate) of the MOSTest procedure through the permutation scheme. In a general sense, we agree that the results of multivariate analysis can be less interpretable in the cases where multiple weakly related phenotypes were selected for joint analysis without proper biological or clinical relevance. However, this is less of an issue in our case, studying regional brain morphology. As we have shown with the brain maps (Figure 2, Supplementary Data 14), the genetic signal of a particular SNP is distributed across the entire brain. Furthermore, in addition to joint analysis of all subcortical and cortical area and thickness measures, we have presented the results from each of these domains in isolation. If needed, one can run analyses more specifically on combining features that belong to e.g. the frontal cortex or any other region of the brain, achieving the desired specificity, and choosing an appropriate trade-off between increased yield versus specificity.

2. In the response, the authors wrote:

"We would further like to mention that an advantage of MOSTest, compared to several other multivariate GWAS methods, e.g. MANOVA, is that it works in two stages: first we compute univariate summary statistics, and then we combine them into a multivariate p-value. Note that the second stage does not require access to raw genotypes. MOSTest can therefore, in principle, be applied in a meta-analytical setting, after combining univariate results across sites with e.g. standard variance-based meta-analysis".

This is not an advantage over other similar methods at all. MANOVA has already been implemented by different people/tools to analyze summary statistics directly, and it has already been applied to GWAS meta-analysis results. I can't see any advantage or novelty as claimed in this statement.

We thank the reviewer for pointing this out. We have now revised the text, stating as a limitation that MOSTest currently require raw genotypes. As noted in the discussion, we are planning to enable MOSTest in a meta-analytical setting, working off summary statistics, while incorporating our permutation scheme to avoid cases with inflated type I error, as we observed in a few simulation cases for MultiABEL results.

“Second, currently, MOSTest requires the availability of raw genotype data. ... Removing these limitations is also a subject of future work.”

3. I asked what the additional associations from multivariate analysis mean in terms of genotype-phenotype map. I don't think this is answered. Let me be more clear, e.g. given the genotypes of the discovered SNPs, can we say anything more about what the phenotypes would look like?

Prediction of phenotypes from genotypes is indeed an important goal, which we plan to address with follow-up studies and further development of MOSTest. It is however beyond the scope of our current analyses; we are not focusing on predicting phenotypes in individuals but on the discovery of the underlying biology, using MOSTest as an omnibus statistical test.

4. I asked an essential question: which traits are driving the multivariate signal? Not answered. Surely the SNPs are detected to be associated with something in the whole group of imaging phenotypes, but which ones? Namely, we don't only want to know that a locus is found for imaging phenotypes, what's going on in the association? The multivariate replication also gives only a p-value, what's being replicated? How do we know the traits driving the signal in discovery samples are the same as those in replication?

Please note that, in response to the previous round of comments, we have added information to Supplementary Data 1-3. There we list, for each significant locus, on which univariate feature(s) there was a significant effect. This is indicated on line 137 of the main manuscript: "Supplementary Data 1-3 lists these loci, together with for how many and which regions the lead SNPs were significant".

We note that MOSTest is an omnibus test, therefore the replication p-value relate to the same null hypothesis that the SNP genotype vector is not associated with any of the input measures. As with the above points, further in-depth investigation of the loci discovered with MOSTest may be achieved by follow-up analyses, but this is out of the scope of our work here.

5. The authors responded:

"Importantly, the multivariate tests show an advantage over minP also when both phenotypes and genotypes are uncorrelated. Therefore, we believe it is incorrect to state that boost in power comes from either phenotypic or genetic correlation."

What does "uncorrelated genotypes" mean? That doesn't sound like genetic correlation. If the two traits are independent, and the SNP affects only one trait, the multivariate test can hardly be more powerful than univariate minP even with Bonferroni correction, as MANOVA simply reduces to Fisher's method. Simple example: two independent traits, significance threshold 0.05, for one trait $p_1 = 0.02 < 0.05/2$ (significant), for the other trait $p_2 = 0.9$ (non-sig.), then Fisher's method multivariate $p = 0.09 > 0.05$ (non-sig.). General comment here, related to questions above: Do not over-sell multivariate analysis.

We apologize for the imprecise formulation in the rebuttal letter. Here, by "uncorrelated phenotypes" and "uncorrelated genotypes" we refer to our simulation scenarios with "re=eye" and "rg=eye", as defined in Supplementary Table 4 "Parameters validated in simulations". In these scenarios, the genetic effects ("beta" in " $y=G \text{ beta} + \text{eps}$ ") are drawn from a normal distribution with identity covariance matrix across phenotypes. Such genetic effects are uncorrelated (i.e. show no genetic correlation), but are not independent (because all traits share the same set of causal SNPs, i.e. SNPs with non-zero beta's). We have shown that, in this scenario, multivariate tests have better statistical power, which is not driven by either phenotypic or genetic correlation, but rather due to polygenic overlap (i.e. shared genetic variance that affects more than one trait).

6. Figure 2: It seems MiXeR is used to estimate the percentage of variance explained by SNPs for a single trait. But in results related to Figure 2, the z-score was transformed from MOSTest which is based on multi-traits. Why can the "transformed z-score" from multi-trait be used for estimating the percentage of explained variance explained for a single trait?

Indeed, the original MiXeR model was developed for p-values calculated by standard univariate GWAS. However, we have shown in Supplementary Figure 8 that the MiXeR model also correctly describes the distribution of p-values produced by MOSTest, including the effects of total linkage disequilibrium and minor allele frequency on the p-values, validating our approach.

Reviewer #3 (Remarks to the Author):

I would like to thank the authors for the clarifications and further analyses/simulations they performed, which makes a very compelling case in favour of using MOSTest for multivariate GWAS using univariate summary statistics. I would like to reiterate that this is a very well written manuscript, and that the MOSTest method could greatly improve SNP discovery. I only have a one minor comment (see below).

Lines 51-54: "While cortical thickness and surface area have been reported to be phenotypically and genetically only weakly correlated to each other, many brain-related traits, e.g. mental disorders, share a large proportion of genetic variants, even in the absence of an overall genetic correlation." To me this still reads as if in there is not genetic correlation between mental disorders, which is not correct. See for example Antilla et al., 2018 (Brainstorm paper), who demonstrated what I would consider to be widespread genetic correlations between mental health diagnoses. Some of those r_G are far from being small/trivial and one must keep in mind that SE may still be large.

Apologies for the lack of clarity. We indicated that cortical thickness and surface area are weakly correlated; there is indeed reasonably high genetic correlation between some pairs of mental disorders while other pairs have a negligible correlation. Our statement intended to convey that many of these genetic correlations do not properly reflect genetic overlap, due to mixed directions of effects. We have now narrowed down our statement as follows:

"We have shown that cortical thickness and surface area have extensive genetic overlap, despite reports that they are phenotypically and genetically only weakly correlated to each other, due to mixed directions of effects of the underlying genetic variants."